# Strength can be controlled by edge dislocations in refractory high-entropy alloys

Chanho Lee[1,2,10], Francesco Maresca[3,4,10 ✉], Rui Feng[1,5,10], Yi Chou[6], T. Ungar[7], Michael Widom [8], Ke An [5], Jonathan D. Poplawsky [9], Yi-Chia Chou [6], Peter K. Liaw [1 ✉] & W. A. Curtin[4]

Energy efficiency is motivating the search for new high-temperature (high-T) metals. Some new body-centered-cubic (BCC) random multicomponent "high-entropy alloys (HEAs)" based on refractory elements (Cr-Mo-Nb-Ta-V-W-Hf-Ti-Zr) possess exceptional strengths at high temperatures but the physical origins of this outstanding behavior are not known. Here we show, using integrated in-situ neutron-diffraction (ND), high-resolution transmission electron microscopy (HRTEM), and recent theory, that the high strength and strength retention of a NbTaTiV alloy and a high-strength/low-density CrMoNbV alloy are attributable to edge dislocations. This finding is surprising because plastic flows in BCC elemental metals and dilute alloys are generally controlled by screw dislocations. We use the insight and theory to perform a computationally-guided search over $10^7$ BCC HEAs and identify over $10^6$ possible ultra-strong high-T alloy compositions for future exploration.

[1] Department of Materials Science and Engineering, The University of Tennessee, Knoxville, TN 37996-2100, USA. [2] Materials Science and Technology Division, Los Alamos National Laboratory, Los Alamos, NM 87545, USA. [3] Engineering and Technology Institute (ENTEG), Faculty of Science and Engineering, University of Groningen, Groningen 9747AG, Netherlands. [4] Laboratory for Multiscale Mechanics Modeling, École Polytechnique Fédérale de Lausanne, CH-1015 Lausanne, Switzerland. [5] Neutron Scattering Division, Oak Ridge National Laboratory, Oak Ridge, TN 37831, USA. [6] Department of Electrophysics, National Yang Ming Chiao Tung University, Hsinchu 30010, Taiwan. [7] Department of Materials Physics, Eötvös University, BudapestP.O. Box 32, H-1518, Hungary. [8] Department of Physics, Carnegie Mellon University, Pittsburgh, PA 15213, USA. [9] Center for Nanophase Materials Sciences, Oak Ridge National Laboratory, Oak Ridge, TN 37831, USA. [10] These authors contributed equally: Chanho Lee, Francesco Maresca, Rui Feng.
✉email: f.maresca@rug.nl; pliaw@utk.edu

Achieving the urgent societal goals of reduced emissions and increasing energy efficiency is driving the development of new materials. One path is lightweight materials (e.g., Mg, Al, and reinforced plastics)[1–3] for low-temperature (T) applications, such as transportation, while a second path is high-temperature (high-T) damage-tolerant materials for increased combustion efficiency (superalloys and TiAl)[4,5]. The new "high-entropy alloys (HEAs)" are single-phase crystalline materials with multiple components that randomly occupy the atomic sites of the crystal lattice[6–11]. HEAs can have remarkable yield strengths, ductility, and/or fracture toughness[10,12,13]. The two body-centered-cubic (BCC) HEAs, MoNbTaW and MoNbTaVW, retain high strengths at temperatures far above those for existing superalloys (Fig. 1)[9,11,14] but the mechanisms enabling this performance are not well-established[11,14]. Recent TEM work has excitingly shown the MoNbTi alloy to deform by dislocations beyond the usually assumed $(110) < 111 > a/2$ screw dislocations[15,16] with few suggestions for alloy discovery.

Here, we show that the high strength and high-T strength retention of both the recent NbTaTiV and new CrMoNbV HEAs (Fig. 1) are controlled by edge dislocations. Our findings are unexpected because screw dislocations are widely understood to control plastic flows in BCC elemental metals and dilute alloys[17,18], although there were hints in the literature that edge dislocation motion might be hindered by solutes in some low/moderate-concentration alloys[19–22]. Unlike in such dilute alloys, our recent theory shows that edge dislocations in some complex HEAs can encounter very large energy barriers to glide[23], and hence, enable high strength that is retained at elevated temperatures. Thus, while the new CrMoNbV alloy reported here has the highest retained strength to date at $T = 1173$ K, a theory-guided search over the entire Al-Cr-Hf-Mo-Nb-Ta-Ti-V-W-Zr composition space predicts over 1,000,000 new alloys with even better performance.

## Results

### Characteristics of NbTaTiV and CrMoNbV HEAs.
Alloys of nominal compositions of NbTaTiV and CrMoNbV were synthesized, and in situ neutron-diffraction (ND) measurements were performed (Supplementary Notes 1 and 2, and Supplementary Fig. 1). The ND patterns show a single BCC solid solution at all temperatures up to 1,173 K (Figs. 2a and 3a). Atom probe tomography reveals NbTaTiV to contain 1.15 at.% (atomic percent) O and 0.45 at.% N, while CrMoNbV has low interstitial contents (0.083 at.% O, 0.034 at.% N, and 0.062 at.% C). The CrMoNbV also has an attractive density ($\rho = 8.08$ g/cm$^3$) with a melting point of $T_m \sim 2502$ K and is studied in the as-cast state. Figure 1 shows the measured yield strengths ($\sigma_y$) versus temperature at a strain rate of $1 \times 10^{-3}$ s$^{-1}$; high strength is retained up to 1173 K (Supplementary Figs. 2 and 3). The strength of NbTaTiV is comparable to MoNbTaW and MoNbTaWV, which have been novel among BCC HEAs for their exceptional high-T strength retention[11,14] and have been predicted to be controlled by edge dislocations[23]. Moreover, the CrMoNbV alloy has the highest reported strength at the homologous temperature of 1173 K[14].

### Determination of type of mobile dislocation via in situ neutron diffraction.
We now demonstrate that edge dislocations, rather than screw dislocations, predominantly control the plasticity in both NbTaTiV and CrMoNbV. Note that screw dislocations can be important in some BCC HEAs[15] but here we highlight that (i) the usually overlooked edge dislocations can control the yield strengths in some BCC HEAs, (ii) unlike screws, edge strengthening can convey high-T strength retention, and (iii) again unlike screws, an accurate parameter-free edge theory exists that enables alloy design. First, the TEM analysis on NbTaTiV shows that the dislocations have Burgers vectors of the <111> a/2 type typical of BCC elements and dilute alloys (Supplementary Fig. 4 and Supplementary Note 3). Second, line broadening of the ND peaks during deformation is due to inhomogeneous strain fields generated by mechanically induced substructures, including dislocations. For a given {hkl} lattice plane in the crystal, ND yields the evolution of both the interplanar spacing, $d_{hkl}$, in the elastic regime and the peak broadening [full width at half maximum, (FWHM), $\Delta d_{hkl}$] in the plastic regime. Figures 2b and 3b present the measured axial and transverse lattice strains versus applied stress at $T = 293$ K. The planar Young's modulus, $E_{hkl}$, and Poisson ratio, $v_{hkl}$, are derived, using the Kroner's model[24], and the cubic elastic constants, $C_{11}$, $C_{12}$, and $C_{44}$, are then computed (Supplementary Fig. 5, Supplementary Table 2, and Supplementary Note 4). The resulting Zener anisotropies, $\frac{2C_{44}}{C_{11}-C_{12}} = 1.2405$ (NbTaTiV) and 0.596 (CrMoNbV), a regime where peak broadening is particularly sensitive to dislocation character (see below). The observed peak broadening during subsequent deformation is analyzed, using the quantitative state-of-the-art Convolutional Multiple Whole Profile (CMWP) method[25]. Note that the CMWP procedure has been widely applied to determine dislocation densities, slip modes, and Burgers vector character in face-centered-cubic (FCC), BCC, and hexagonal-close-packed (HCP) crystals[25,26]. In addition to the instrumental response, HEAs have a peak broadening in the undeformed state due to local deviations of atoms from the perfect lattice positions. It is important that this initial broadening be subtracted from the mean-square broadening at finite deformation to isolate the role of the dislocations. The line broadening by dislocations is related to the dislocation contrast factor for each {hkl} plane via the q parameter as

$$C_{hkl} = C_{h00}\left[1 - q\left(\frac{h^2k^2 + k^2l^2 + h^2l^2}{h^2 + k^2 + l^2}\right)\right] \quad (1)$$

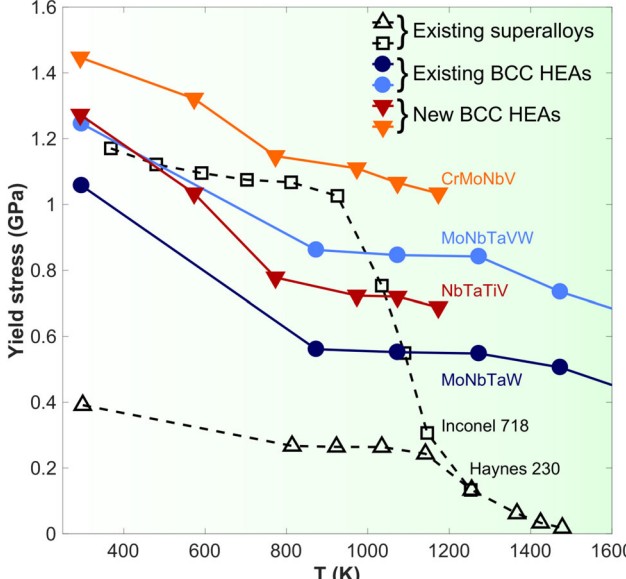

**Fig. 1 Alloy strength vs. temperature.** The refractory BCC HEAs retain high strengths up to temperatures well beyond those where superalloys lose strengths. The NbTaTiV alloy here is comparable to the literature MoNbTaVW and MoNbTaW alloys. The strength of the CrMoNbV alloy, predicted to be even stronger, significantly exceeds the strengths of the existing BCC HEAs. The strengths of NbTaTiV and CrMoNbV are found here to be controlled by edge dislocations—see Figs. 2, 3, and 4—not screw dislocations. Literature data are reproduced from ref. [8].

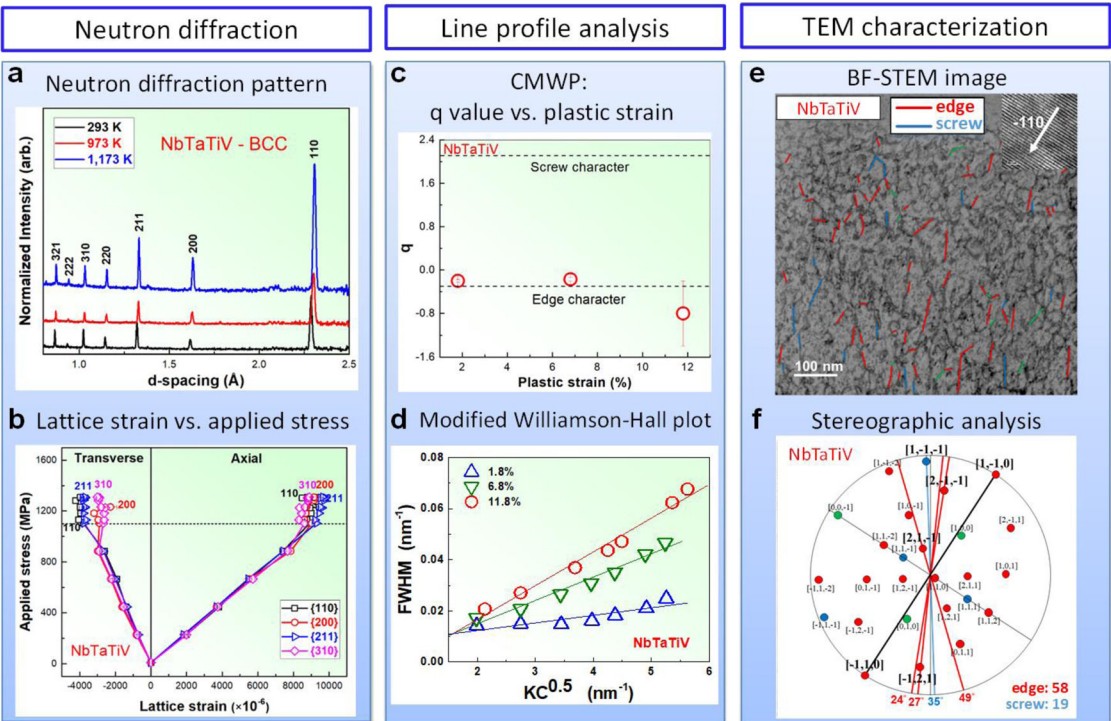

**Fig. 2 Neutron diffraction and TEM experiments showing the dominance of edge dislocations in NbTaTiV. a** Neutron-diffraction patterns presenting interplanar spacings with peaks indexed for the BCC structure of NbTaTiV, at temperatures of 293, 973, and 1173 K. **b** Axial and transverse lattice strains versus applied load, shown as the applied stress versus lattice strain so that the slopes correspond to the planar Young's moduli ($E_{hkl}$) and Young's moduli/ Poisson's ratios ($E_{hkl}/\nu_{hkl}$), as indicated for the {110}, {200}, {211}, and {310} planes, respectively, at $T = 293$ K. The onset of plastic yielding (yield stress) is presented as the dashed line. **c** Evolution of q parameters as a function of plastic strain, which is obtained from the Convolutional Multiple Whole Profile (CMWP) fitting. Dashed lines indicate values of q parameters for edge and screw character dislocations, considering a 15 % error margin for the elastic constants. The error bars in the q-parameter values are determined by the fractions of edge or screw dislocations. **d** Modified-Williamson-Hall plot, FWHM versus $KC^{0.5}$ at plastic strains of 1.8 %, 6.8 %, and 11.8 %, respectively, at $T = 293$ K. The plots were obtained from the physical profiles calculated by the CMWP procedure, considering free of the instrumental effects on the FWHM data. The pattern of the undeformed specimen was applied to the CMWP procedure as an instrumental pattern. The much better agreement of the data with the edge analysis demonstrates th**e** dominance of edge dislocations. **e** Annular-Bright-field (ABF)-STEM image of NbTaTiV at a plastic strain of 11.8% with a two-beam condition near $Z = [110]$ and $\vec{g} = (\bar{1}10)$. All straight dislocation lines longer than 5 nm are highlighted by blue and red lines, corresponding to their identification as edge and screw dislocations, respectively. **f** Stereographic projection related to the [110] orientation, where [$\bar{1}$10] has been aligned with images in (**e**). All possible dislocations are indicated, and those corresponding to the images in (**e**) are highlighted in bold. The degrees indicate the angles with respect to the [$\bar{1}$10] direction. Blue lines/blue symbols present those dislocations identified as edge, and red lines/symbols indicate those dislocations identified as screw. The green lines/symbols represented the dislocations that could be either edge or screw, which was excluded in the ratio calculation. As a result, ~75% of the dislocations are identified as edge.

where $q$ is a function of only the dislocation Burgers vector, character, and elastic anisotropy. The CMWP analysis performed on NbTaTiV samples, for <111>a/2 dislocations and deformed to plastic strains of 1.8, 6.8, and 11.8%, reveals $q$ values, as shown in Fig. 2c. These are all consistent with the theoretical value of $-0.6 < q < -0.4$ (accounting for uncertainty in the elastic constants) for edge dislocations, and far differ in both the magnitude and sign than the range of $+1.9 < q < +2.1$ for screw dislocations. The simpler modified Williamson-Hall (mWH) analysis[16,27] of $\Delta K_{hkl} = -\frac{\Delta d_{hkl}}{d_{hkl}^2}$ versus $K = \frac{1}{d}$ as $(\Delta K_{hkl})^2 = (0.9/D)^2 + BK_{hkl}^2 C_{hkl}$ ($B$ a constant) yields the same conclusion (Fig. 2d). Similarly, CMWP and mWH analyses for the CrMoNbV alloy at plastic strains of 0.8, 4.2, and 5.3 % yield $q$ values presented in Fig. 3c,d, which can be compared to the theoretical edge value of $-2$ and the theoretical screw value of $+1.5$. There are larger errors in the q-parameter values, which determine the fractions of dislocations with screw or edge character. However, the general conclusions about the overwhelming edge character are not influenced by the experimental uncertainty. The quality of the

neutron pattern for CrMoNbV is lower than for NbTaTiV, especially at the very low plastic strain of 0.8%, making that result less reliable. Overall, the neutron data demonstrate that edge dislocations are more dominant than screw in both NbTaTiV and CrMoNbV HEAs.

**Determination of type of mobile dislocation via BF-STEM and stereographic analysis.** The significant role of edge dislocations in NbTaTiV and CrMoNbV is further supported by the bright-field (BF)-scanning transmission electron microscopy (STEM) and Stereographic analysis of specimens deformed at $T = 293$ K. The dislocation core structures and diffraction pattern of strained NbTaTiV and CrMoNbV are displayed in Supplementary Fig. 6 in the Supplementary Information (Supplementary Note 5). Figures 2e and 3e show the BF-STEM images of the dislocation networks in NbTaTiV and CrMoNbV, at plastic strains of 11.8% and 4.2%, respectively, taken slightly off the [110] zone axis to enhance the contrast. All dislocations having a line of length over 5 nm are identified (blue and red lines). The identified line

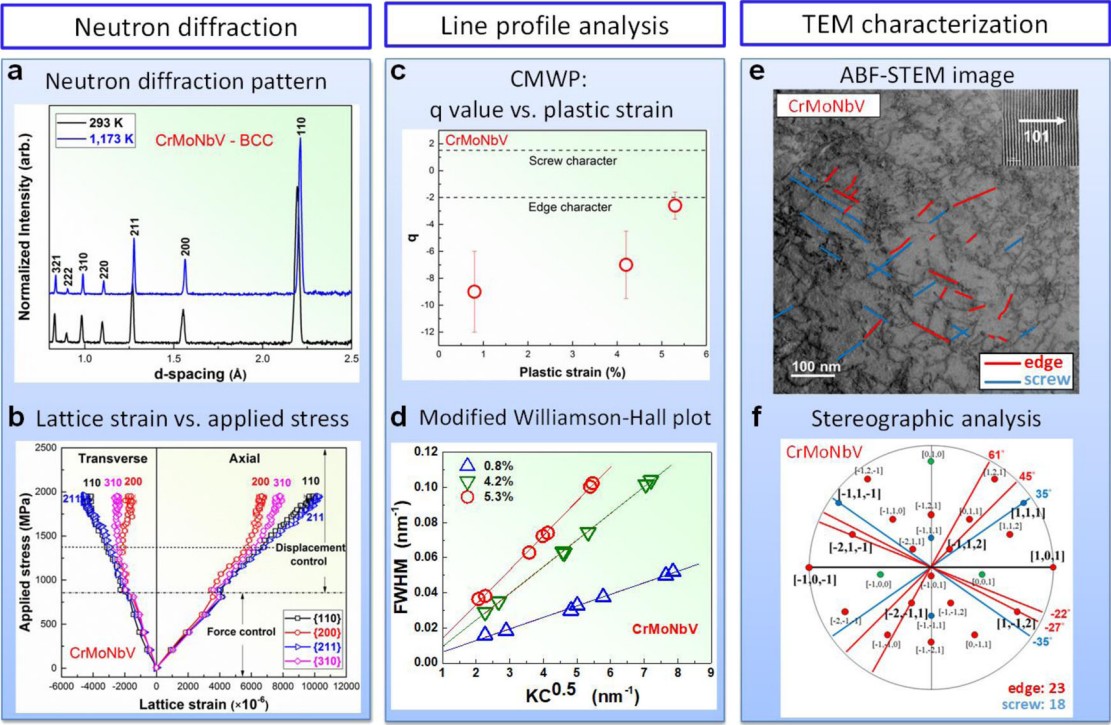

**Fig. 3 Neutron diffraction and TEM experiments showing the dominance of edge dislocations in CrMoNbV. a** Neutron-diffraction patterns presenting interplanar spacings with peaks indexed for the BCC structure of CrMoNbV, at temperatures of 293 and 1173 K. **b** Axial and transverse lattice strains versus applied load, shown as the applied stress versus lattice strain so that the slopes correspond to the planar Young's moduli ($E_{hkl}$) and Young's moduli/ Poisson's ratios ($E_{hkl}/\nu_{hkl}$), as indicated for the {110}, {200}, {211}, and {310} planes, respectively, at $T = 293$ K. The onset of plastic yielding (yield stress) is presented as the dashed line. **c** Evolution of q parameters as a function of plastic strain, which is obtained from the Convolutional Multiple Whole Profile (CMWP) fitting. Dashed lines indicate values of q parameters for edge and screw character dislocations, considering a 15% error margin for the elastic constants. The error bars in the q-parameter values are determined by the fractions of edge or screw dislocations. **d** Modified-Williamson-Hall plot, FWHM versus $KC^{0.5}$ at plastic strains of 0.8%, 4.2%, and 5.3%, respectively, at $T = 293$ K. The plots were obtained from the physical profiles calculated by the CMWP procedure, considering free of the instrumental effects on the FWHM data. The pattern of the undeformed specimen was applied to the CMWP procedure as an instrumental pattern. The much better agreement of the data with the edge analysis demonstrates the **e** dominance of edge dislocations. **e** Annular-Bright-field (ABF)-STEM image of CrMoNbV at a plastic strain of 4.2% with a two-beam condition near $Z = [\bar{1}01]$ and $\vec{g} = (101)$. All straight dislocation lines longer than 5 nm are highlighted by blue and red lines, corresponding to their identifications as edge and screw dislocations, respectively. **f** Stereographic projection related to the $[\bar{1}01]$ orientation, where [101] has been aligned with images in (**e**). All possible dislocations are indicated, and those corresponding to the images in (**e**) are highlighted in bold. The degrees indicate the angles with respect to the [101] direction. Blue lines/blue symbols present those dislocations identified as edge, and red lines/symbols indicate those dislocations identified as screw. The green lines/symbols represented the dislocations that could be either edge or screw, which was excluded in the ratio calculation. As a result, ~55% of the dislocations are identified as edge.

direction is compared with the stereographic projection. If the difference is <5 degrees, then the dislocation character is determined. If two characters are possible, the one with the closest match to the dislocation line is chosen. If the dislocation line does not meet these criteria, it is not considered (e.g., the finer-scale dislocation tangles in the images). Figures 2f and 3f present the results of the stereographic-projection analysis, which reveals the characters of the long straight dislocation lines. Note that <111>a/ 2 dislocations on {110} and {112} slip planes were frequently observed in NbTaTiV and CrMoNbV, and we found the same type of slip systems with previously studied work by Wang et al.[15]. The observed dislocations (blue and red in Figs. 2e and 3e) are highlighted in bold and lie along the lines presented. In NbTaTiV, the measured dislocations are predominantly of edge character (58 of 77, or 75%). In CrMoNbV, the dislocations are again predominantly edge (23 of 41, or 56%). Dislocation contrast, i.e., the **g·b** analysis, can also reveal dislocation character for some cases. The analysis described in Supplementary Note 3 further demonstrates the dominance of edge dislocations in NbTaTiV. The characterizations of dislocations in the high-T-

deformed NbTaTiV and CrMoNbV alloys are further investigated via the BF-STEM and Stereographic analysis of specimens deformed at $T = 1,173$ K. (Supplementary Fig. 7) It is found that the {110}<111> edge dislocations with the <112> dislocation line can be widely observed in NbTaTiV and CrMoNbV. The measured dislocations are predominantly of edge character for both HEAs (27 of 33, or 82% in NbTaTiV and 81 of 127, or 64% in CrMoNbV), indicating the maintenance of edge-dislocation-dominant deformation at 1173 K.

The observation of a high fraction of edge dislocations relative to screw dislocations is in distinct contrast to typical studies in BCC metals and other HEAs[28] that almost exclusively show long straight screw dislocations and do not have good high-T strengths. While results from TEM and neutron diffraction are not quantitatively identical, both results consistently show that edge dislocations are prevalent and dominant, as compared to screws. Our experimental demonstration that plasticity in a BCC alloy can be controlled by edge dislocations is very surprising because it is counter to the nearly universally accepted understanding, based on many observations in elemental and dilute

BCC alloys over many decades, that screw dislocations completely dominate the plastic-flow behavior[17,18].

## Discussion

**Theory and predictions for role of edge dislocations in the strengthening.** To further cement that the edge dislocations can control the strength, we apply a new theory for the yield strength versus temperature and strain rate for edge dislocations moving through a random BCC alloy[23]. In the theory, each alloying element is viewed as a solute that interacts with a dislocation in a hypothetical homogeneous "average" alloy that has all the macroscopic properties of the true random alloy. Fluctuations in the local arrangements of solutes create large local variations in the potential energy for the dislocation. The dislocation thus spontaneously adopts a low-energy wavy structure to take advantage of the low-energy solute environments and avoid the high-energy environments. The plastic flow then requires the temperature- and stress-assisted thermal activation of the dislocations out of the low-energy environments and over the large barriers created by the adjacent high-energy environments along the glide plane in the random alloy. The full theory is reduced to an analytic model, using the elasticity approximation, $U^i(x, y) = -p(x, y)\Delta V_i$, for the solute/dislocation interaction, $U^i(x, y)$, of a solute of the type, $i$, at a position, $(x, y)$, under the pressure field, $p(x,y)$, due to a dislocation lying along $z$ and centered at the origin, where $\Delta V_i$ is the misfit volume of the type-$i$ solute in the average alloy. The yield stress as a function of temperature and strain rate, $\dot{\varepsilon}$, is then[23]

$$\sigma_y(T, \dot{\varepsilon}) = \sigma_{y0}\left[1 - \left(\frac{kT}{\Delta E_{b0}}\ln\frac{\dot{\varepsilon}_0}{\dot{\varepsilon}}\right)^{2/3}\right], \frac{\sigma_y}{\sigma_{y0}} \geq 0.5 \quad (2)$$

$$\sigma_y(T, \dot{\varepsilon}) = \sigma_{y0}\exp\left(-\frac{1}{0.55}\frac{kT}{\Delta E_{b0}}\ln\frac{\varepsilon_0}{\varepsilon}\right), \frac{\sigma_y}{\sigma_{y0}} < 0.5 \quad (3)$$

where the zero-temperature yield stress, $\sigma_{y0}$, and zero-stress energy barrier, $\Delta E_{b0}$, are computed as

$$\sigma_{y0} = 3.067 A_\sigma \alpha^{-1/3}\mu\left(\frac{1+\nu}{1-\nu}\right)^{4/3}\left(\sum_i \frac{c_i \Delta V_i^2}{b^6}\right)^{2/3} \quad (4)$$

$$\Delta E_{b0} = A_E \alpha^{1/3}\mu b^3\left(\frac{1+\nu}{1-\nu}\right)^{2/3}\left(\sum_i \frac{c_i \Delta V_i^2}{b^6}\right)^{1/3} \quad (5)$$

Here, $\mu$ and $\nu$ are the isotropic alloy elastic constants, $\{c_i\}$ the solute concentrations, $\alpha = 1/12$ a fixed line tension coefficient, and $\dot{\varepsilon}_0 = 10^4 s^{-1}$ a reference strain rate. Predictions at high-T are insensitive to the line-tension coefficient, and room-temperature (RT) predictions vary modestly with this coefficient. The coefficients, $A_\sigma$ and $A_E$, are computed in the full theory for each alloy composition. There are thus no fitting parameters in the theory, only material properties (elastic constants, misfit volumes, and line tension) and alloy composition. However, the coefficients fall in a narrow range of $A_\sigma = 0.040 \pm 0.004$, and $A_E = 2.00 \pm 0.2$ across a wide spectrum of BCC HEAs, indicating that the dominant material properties are the misfit volumes and elastic constants.

Applying the theory to NbTaTiV, we first compute the misfit volumes of all solutes using Vegard's Law, which accurately predicts alloy atomic volumes of refractory BCC HEAs studied to date[9]. The BCC atomic volumes, $\{V_{0i}\}$, are Nb = 17.952, Ta = 17.985, Ti = 17.387, and V = 14.02 in Å$^3$ (see the Supplementary Note 7). The alloy volume is $V = \sum_i c_i V_{0i}$, and the misfit volume for the solute, $i$, is then $\Delta V_i = V_{0i} - V$. The alloy elastic constants are computed, employing the rule of mixtures for the elemental $C_{11}$, $C_{12}$, and $C_{44}$ to obtain the bulk

modulus, $B = \frac{C_{11}+2C_{12}}{3}$, and $\mu = \sqrt{C_{44}(C_{11} - C_{12})/2} = 47.8\,\text{GPa}$, and thus, $\nu = \frac{3B-2\mu}{2(3B+\mu)} = 0.365$, all consistent with experiments (Supplementary Fig. 5). We use the precisely computed coefficients, $A_\sigma = 0.0437$, and $A_E = 2.02$, from the full theory for the NbTaTiV alloy. The predicted strength versus temperature for NbTaTiV with no interstitial content at the experimental strain rate of $10^{-3}$/s is shown in Fig. 4a along with the present experiments and literature data. The theory trend is good but much lower than the present experiments, although comparable to the literature data at $T = 300$ K. We attribute this difference to the 1.6 atomic percent (at. %) O + N impurities in our alloy because it is known that 2 at% O or N strengthens a BCC HEA by ~400–500 MPa at $T = 293$ K (Fig. 4a)[29]. Furthermore, we have computed the tetragonal misfit strains for O and N in Nb, Ta, and V, via density functional theory (DFT), and they are all large ($\varepsilon_{11} \sim 0.60$; $\varepsilon_{22} = \varepsilon_{33} \sim -0.1$), consistent with high strengthening. Semi-quantitatively, using a new Nb-O interatomic potential[30] to compute the interaction of O with an edge dislocation in Nb and assuming the same interaction energies for the NbTaVTi alloy, the theory can be extended to include the addition of 1.6 at.% O interstitials and predicts an increase in strength of ~300 MPa at $T = 300$ K, reaching a reasonable agreement with the experiment (Fig. 4a).

We next apply the theory to the CrMoNbV alloy. Comparison with experiments is facilitated by the very low interstitial content, as compared to NbTaTiV. Equations 4 and 5 show that the solute-misfit volumes are the critical component for strengthening, and Cr has a small BCC atomic volume of 12.321 Å$^3$ and so should lead to high strengths. We apply Eqs. 2–5 using (i) Vegard's law, (ii) the additional atomic volumes of Cr = 12.321 Å$^3$, and Mo = 15.524 Å$^3$ (see Supplementary Note 7), (iii) alloy elastic moduli of $\mu = 76.6$ GPa, and $\nu = 0.336$, and (iv) the precisely computed coefficients, $A_\sigma = 0.0344$ and $A_E = 1.93$, for CrMoNbV. The predicted yield strength versus temperature for CrMoNbV is shown in Fig. 4a and, with no fitting, broadly agrees with the measured strengths. The RT prediction may be higher due to the line tension coefficient[21] or microfractures in the alloy. As discussed previously, this model also does not capture the plateau at 973–1273 K that is observed in many BCC HEAs[21] but has been shown to capture the further strength decreases found in other alloys at higher temperatures. The predicted and measured strengths of this alloy exceed those of all previously reported single-phase BCC HEA alloys at the temperature of 1173 K[11,14,31] and at the homologous temperature of $T/T_m = 0.47$[14] (e.g., Fig. 1). Our predictions based on the edge dislocation are fully consistent with their significant presence, as revealed by both neutron diffraction and BF-STEM.

With the understanding of the key role of edge dislocations in the strengthening of BCC HEAs, especially at high temperatures, we can now identify new promising alloy compositions. Screw dislocations remain important in some BCC HEAs, but the vacancy-driven unpinning of the high-strength screw cross-kinks is expected to lead to a loss of strength at high T[31]. Moreover, accurate inputs for screw theories are difficult to obtain[32,33], and hence, a computationally guided search for new high-performance alloys based on edge-dislocation strengthening is a computable mechanistic path for the design and discovery of further new alloys with high strengths and high-T strength retention, independent of whether or not screw dislocations provide strengthening. For the rapid alloy design, we use Eqs. 2–5, the average coefficients of $A_\sigma = 0.040$ and $A_E = 2.00$, Vegard's Law, and the rule-of-mixtures for elastic constants, to search across more than 10,000,000 compositions in the 10-component Al-Cr-Hf-Mo-Nb-Ta-Ti-V-W-Zr family (the Supplementary Note 7, including a MATLAB code, with the data publicly available[34]).

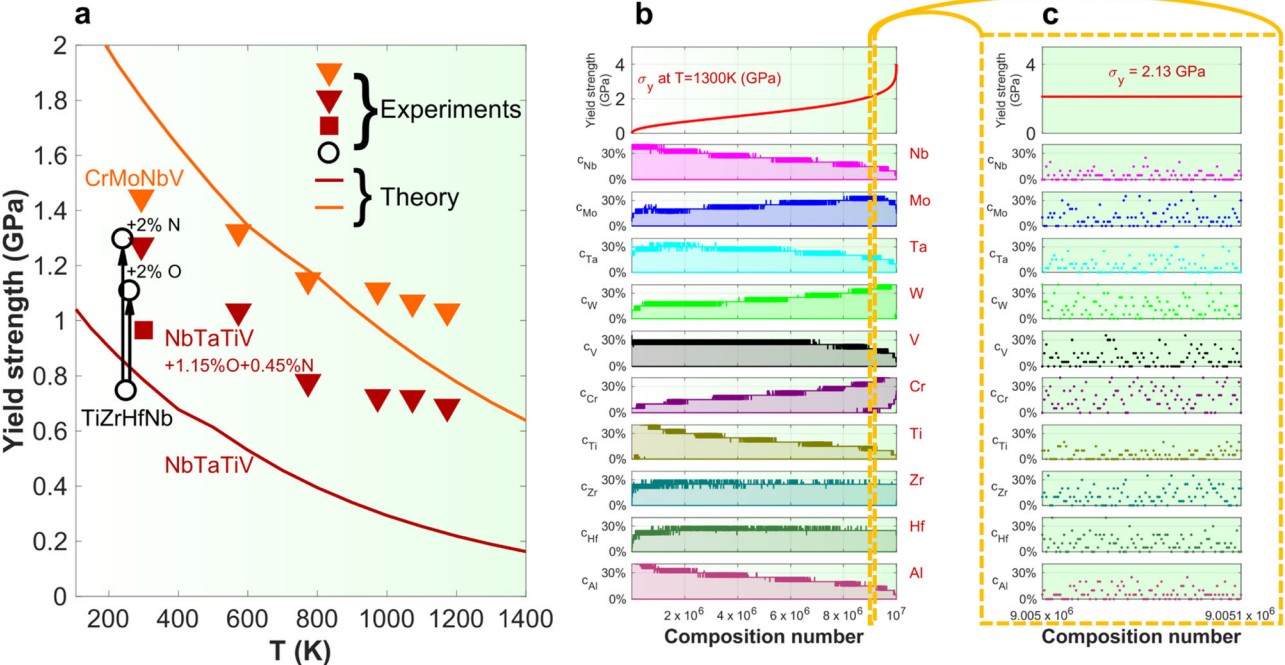

**Fig. 4 Theory predictions of yield strengths in BCC HEAs. a** Yield strength vs. temperature, experiments and theory for NbTaTiV and CrMoNbV. NbTaTiV: Experiments on the NbTaTiV alloy with 1.15 at.% O and 0.45 at.% N: red triangles at a strain rate of $10^{-3}\,\mathrm{s^{-1}}$; theory for an interstitial-free alloy: red line. Red square reported in ref. [9] at $5 \times 10^{-4}\,\mathrm{s^{-1}}$ with no O or N impurity content reported. Black circles: experiments on TiHfZrNb at $T = 300\,\mathrm{K}$ with and without 2 at.% O and N that show strength increases up to ~500 MPa[21], comparable to the difference here between model predictions for the interstitial-free NbTaTiV and the experiments with 1.6 at.% impurities. CrMoNbV: Experiments: orange triangles; theory: orange line; no impurities are detected in this alloy. The strength of the new CrMoNbV at 1173 K exceeds those of all previously reported alloys. **b** Theory predictions for $T = 1300\,\mathrm{K}$ strength vs. composition. As detailed in Supplementary Note 7, computations of the yield strengths have been performed for >10,000,000 alloys. The alloys have been assigned with an increasing number as a function of the increasing strength. Thus, the lowest strength alloy is the number 1, and the highest strength alloy is the number 10,003,049. For better visualization, we have grouped the alloys into bins containing 1000 compositions each (see Supplementary Note 7). The lowest compositions are for alloys 1 to 1000 and enter the 1st bin. The second bin contains alloys 1001 to 2000, etc. Within each bin, the interval between the 10th and the 90th percentiles of the elemental contents (Nb, Mo, Ta, etc.) is computed. In this panel, the shaded areas indicate the concentrations between the 10th and the 90th percentiles, evaluated over the compositional bins. Thus, the y axis is the elemental concentration. The screening includes >10,000,000 compositions in the Nb-Mo-Ta-W-V-Cr-Ti-Zr-Hf-Al compositional space. **c** A zoom-in of the screening for 100 compositions around the 1,000,000 strongest alloys (between composition numbers of 9,005,000 and 9,005,100). Here, the actual alloy contents per composition number are shown. There are plenty of possible alloys that can have the same yield strength, but different compositions.

Figure [4]b and c (see the Supplementary Note 7 for more details) show ~6,000,000 alloys with estimated strengths over 1 GPa at $T = 1300\,\mathrm{K}$, and ~1,300,000 over 2 GPa, far exceeding the strengths of any existing alloys. At 1300 K, many alloys also have strength/density >0.25 GPa g/cm³ (see Supplementary Note 7) that is the highest achieved to date *at* room temperature. We propose two super-high-strength/high-T alloys of Mo₅W₂.₅CrZrHf and Mo₂.₅W₂.₅CrZrHf for fabrication and testing, with predicted strengths of ~3 GPa at 1300 K. Preliminary thermodynamic assessment (Supplementary Note 8) suggests the use of low concentrations of Cr, Zr, and Hf to avoid intermetallic formation. We also propose two super-high-strength/high-T alloys, Mo₆WCrZrHf and Mo₂.₅TaWV₂.₅CrZrHf with lower W, having predicted strengths of ~2.5 GPa at 1300 K. Future combinations of our design strategy with detailed thermodynamics[35,36] and added constraints (e.g., high ductility criterion) may lead to the discovery of new alloys that can achieve the multi-objective performance required in many critical engineering applications.

To conclude, the present work, motivated by recent theory, demonstrates by means of an ample array of experimental techniques that edge dislocations can control the strengths of BCC high-entropy alloys. This finding supports using a mechanistic theory based on edge-dislocation strengthening in BCC alloys to search for new, strong alloys. As shown in the current work, this theory is a validated and viable way to perform the combinatorial search in the immense HEAs compositional space.

## Methods

**Materials preparations and microstructural characterization.** The NbTaTiV alloys were manufactured by arc-melting under an argon atmosphere on a water-cooled Cu hearth from Nb, Ta, Ti, and V elements of 99.99 weight percent (wt. %) purity. The nominal composition of the present alloy is Nb₂₅Ta₂₅Ti₂₅V₂₅ in atomic percent (at.%). To ensure a full synthesis of composed elements, the ingot was melted over 10 times. From the master alloys, the specimens were fabricated, followed by direct casting into cylindrical rods with a 4-mm diameter and 50-mm length, using a drop-casting technique. Similarly, the CrMoNbV alloy was manufactured by arc-melting the constituent elements of Cr, Mo, Nb, and V (purity 99.9 wt.%), followed by drop casting into a water-cooled copper hearth. The NbTaTiV alloys were sealed with triple-pumped argon in quartz tubes and homogenized at 1473 K for 3 days, followed by water cooling. The CrMoNbV samples were tested and characterized in the as-cast state due to the very high homogenous temperature (>1673 K). The microstructure was examined by scanning electron microscopy (SEM), using a Zeiss Auriga 40 equipped with back-scattered electrons (BSE) detector. The chemical composition of the alloy was studied by atom probe tomography (APT) analysis after the homogenization treatment. APT experiments were conducted in a CAMECA LEAP 4000XHR using laser mode with a 50 pJ laser energy, 30 K base temperature, a 200 kHz pulse repetition rate, and a 0.5% detection rate.

**Mechanical tests.** The mechanical tests were performed under uniaxial compression at elevated temperatures, employing a computer-controlled Materials Testing System (MTS) servo-hydraulic-testing machine. Tests on NbTaTiV were

conducted at an initial strain rate of $1 \times 10^{-3}$ s$^{-1}$ on specimens of 4-mm diameter and 8-mm length. Tests on CrMoNbV were performed at a strain rate of $2 \times 10^{-4}$ s$^{-1}$ on specimens of 3-mm diameter and 6-mm length. Test samples were heated to and held at the desired temperatures for at least 30 min. until the temperature was stabilized within ±10 K.

**In situ ND experiments**. The in situ ND instrument utilizes time-of-flight (TOF) measurements, which allows covering a wide range of d spacings without the rotation of detectors or samples. Two detectors at ±90 degrees are used to collect diffracted beams from the polycrystalline grains with lattice planes parallel to the axial and transverse directions, respectively. The compression experiments on the homogenization-treated NbTaTiV alloy with a diameter of 4 mm and length of 8 mm, and the as-cast CrMoNbV alloy with a diameter of 4.5 mm and length of 9 mm, were conducted to investigate the elastic and plastic deformation behaviors at room and elevated temperatures, using an MTS load frame. The samples were illuminated by the incident neutron beam of a $3 \times 3$ mm$^2$ slit size and 2-mm receiving collimators. During the in-situ measurement of the diffraction patterns for the elastic-deformation period, a stepwise-force control sequence was utilized. At each stress level, the measurement times of the ND data were 20 and 12 min for NbTaTiV and CrMoNbV, respectively. For NbTaTiV, when the stress level reached 1100 MPa for 293 K, 750 MPa for 973 K, and 620 MPa for 1173 K (close to the macroscopic yield strength), a stepwise-displacement control with an incremental step of 0.2 mm was employed. Similarly, for the CrMoNbV alloy, a stepwise displacement control with a displacement rate of $1.5 \times 10^{-4}$ mm/s was also used when the stress exceeded 800 MPa. The collected data were analyzed by single-peak fitting, using the VULCAN Data Reduction and Interactive Visualization software (VDRIVE) program[37].

**Characterization of dislocations via TEM and STEM**. The transmission electron microscopy (TEM) and scanning STEM images of NbTaTiV and CrMoNbV were taken and analyzed, using a JEOL ARM200F TEM/STEM with spherical aberration correctors. Stereographic projections of dislocations were used to identify the dislocation types and orientations, as discussed in Supplementary Note 3.

## Data availability
The public repository with the raw data and a MATLAB code to predict the yield strengths of BCC HEAs as a function of temperature generated in this study are provided in https://archive.materialscloud.org/record/2021.65 and https://doi.org/10.24435/materialscloud:fs-27.

## Code availability
The MATLAB code included as a separate Supplementary Data 1.

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

## Acknowledgements
The authors thank Michael C. Gao for help in the design of the single-phase refractory HEAs using the CALculation of PHAse Diagram (CALPHAD) and Gian Song and Hahn Choo for help with the analysis of the in-situ neutron diffraction data. The authors thank Dr. Binglun Yin for providing the DFT computation of the O and N misfit strains in Nb, Ta, and V. The authors thank Mr. Xuesong Fan and Mr. Hugh Shortt for their

experimental efforts during the revision processes. C.L., R.F., and P.K.L. thank the U.S. Army Research Office for the support of the present work through projects, W911NF-13-1-0438 and W911NF-19-2-0049. P.K.L. thanks the National Science Foundation for the support of the present work through projects, DMR-1611180 and 1809640. C.L. acknowledges the partial support from the Center of Materials Processing, a Tennessee Higher Education Commission (THEC) Center of Excellence located at The University of Tennessee, Knoxville. F.M. and W.A.C. acknowledge the partial support for the current work from the European Research Commission Advanced Grant, "Predictive Computational Metallurgy", ERC Grant agreement No. 339081 - PreCoMet. F.M. and W.A.C. also thank Prof. H. Sheng and Prof. E. Ma for sharing their Nb-O interatomic potential. Y.C.C. acknowledges the support from the Ministry of Science and Technology of Taiwan (MOST) under Grant No. MOST-110-2636-M-009-008. All authors acknowledge the core facility support at NCTU from MOST. The present work was partially supported by the "Center for the Semiconductor Technology Research" from The Featured Areas Research Center Program within the framework of the Higher Education Sprout Project by the Ministry of Education (MOE) in Taiwan and supported in part by the Ministry of Science and Technology, Taiwan, under Grant No. MOST 110-2634-F-009-027. Research at the Spallation Neutron Source (SNS), the Oak Ridge National Laboratory (ORNL), was partially sponsored by the Scientific User Facilities Division, Office of Basic Energy Sciences, U.S. Department of Energy (DOE). R.F. thanks for the support from the Materials and Engineering Initiative at SNS, ORNL. The atom probe tomography (APT) experiments were performed at ORNL's Center for the Nanophase Materials Science (CNMS), which is a DOE Office of Science User Facility. Thermodynamic stability modeling was supported by the DOE under grant DE-SC0014506

## Author contributions

C.L., R.F., K.A., and P.K.L. performed the in situ neutron experiments and data analyses. Y.C., and Y.-C.C. carried out the Bright-Field (BF)-Scanning Transmission Electron Microscopy (STEM) and Stereographic analysis. J.D.P. performed the atom-probe-tomography (APT) experiments, data analysis, and interpretation. T.U. contributed the Williamson-Hall (W-H) profile modeling. F.M., M.W., and W.A.C. developed the theory and predictions to identify the role of edge dislocations in strengthening. All authors wrote manuscript together and contributed to the discussions and revisions of the paper.

## Competing interests

The authors declare no competing interests.
