## [Peer Review File · Nature Communications]

Strength Can be Controlled by Edge Dislocations in Refractory High-Entropy AlloysREVIEWER COMMENTS

Reviewer #1 (Remarks to the Author):

This manuscript reports temperature dependence of yield strengths of BCC HEAs: NbTaTiV and CrMoNbV, and attributes the high strength and high strength retention to edge dislocations. Through a computer search a million new alloy compositions are identified “for future exploration”.

- In BCC metals, screw dislocations have non-planar core structure and hence, exhibit lower mobility as compared to edge dislocation. However, this is primarily true at low homologous temperatures. The elevated temperature strengths ($> 0.5T_m$), where thermal activation can overcome the Peierls barrier, cannot be explained based on low mobility of screw dislocations at low homologous temperatures. Hence, the authors claim that strength retention at elevated temperatures can be attributed to edge dislocations appears somewhat confusing.

- The data in Figure 1 show three stages: steep increase in strength with decreasing temperatures at lower temperatures, near plateau at intermediate temperatures and gradual decline at elevated temperatures. It is not clear that a single mechanism can explain the entire temperature range. The model predictions (Fig 4) show a poor match with the experimental data.

- The TEM characterization (Fig 2 and 3) only shows room temperature deformed samples. Also, no information is provided on the core structure of the dislocations either by atomistic modeling or by HRTEM.

- In Fig. 1, comparison of yield strengths for different materials would be more meaningful if the strength values were normalized by modulus.

- References #16 and #25 appear to be the same.

Reviewer #2 (Remarks to the Author):

In this manuscript, the authors present experimental and theoretical arguments to show that the yield stress of refractory HEA is controlled by the glide of edge rather than screw dislocations. Based on this, the authors scan a large composition range of refractory HEA and propose new compositions with expected outstanding strength at high temperatures.

The arguments proposed by the authors show that edge dislocations are relevant and should not be discarded - in contrast with pure BCC metals where the kink-pair nucleation on screw dislocation is the plasticity-controlling process. Based on the points listed below, I will argue that the results presented in this manuscript are however not strong enough to discard the role of screw dislocations in these alloys, which appears as an implicit assumption to apply the edge-based theory to explore compositions as performed in the last part of the paper. In addition, a recent publication (Ref. [15] of the present manuscript) shows the importance of dislocation glide on higher order glide systems. Even if this reference is cited in the manuscript, the results reported here are not discussed in regard of these findings. Please find below a tailed list of comments that need to be addressed:

- l. 50-53: These statements need to be discussed in regards of the findings of Ref. [15] .
- l. 54-83: I am not an expert of the CMWP method. How reliable is this technique and has it been applied successfully to other metal and alloys ? Is there a reference where this method has been applied to BCC metals and BCC dilute alloys and demonstrate that the plasticity is controlled by screw dislocations ? Indeed, the large error bars and the results obtained in the case of CrMoNbV (finding $q = -10$ instead of belonging to $[-2, 2]$) appears as worrisome to a naive reader.
- l. 84-100: The STEM results presented here do not appear very conclusive. Looking at Fig. 2.e and 2.f, it seems that the microstructures of both alloys display long screw segments, at least in comparable proportion than edge dislocations. How is obtained the 75% and the 56% of edge component discussed in the manuscript ? Aren't these numbers weighted by dislocation length ?
- l. 84-100: These results should be discussed in light of the findings of Ref. [15]. Did the author find evidence of other slip systems for these alloys ?
- Fig. 2.e: green segments appear in Fig. 2.e and it is not explained what they represent.
- Fig. 2.d and 3.d: the label of the x-axes are inconsistent.
- l. 110-158: the applicability of the edge model relies on the assumption that the role of the screw dislocations can be discarded. Because of the above comments, the experimental proof do not appear convincing enough to discard the role of screw dislocations in these alloys. Moreover, the findings of ref.[15] suggest the important role of other slip system. Could this model be applied to other slip system in order to clarify their roles ?
- Fig. 4: The comparison of the red theoretical curve with experimental points for NbTaTiV accounting for interstitials appears reasonable. However the comparison of experimental points with the theoretical curve for the CrMoNbV alloy appears less convincing. Indeed, while the theory gives satisfactory results for intermediate temperatures, it seems that the global temperature dependence of the yield stress is not reproduced by the model: the model overestimates (underestimates) the yield stress at low (high) T respectively. This tendency is not discussed in the manuscript and might suggest

that the model does not include all the possible mechanisms (as the role screw dislocation and other slip systems) that may play an important role at low or high temperature.

- Even if we concede that the plasticity of CrMoNbV and NbTaTiV are controlled by edge dislocations, it may not be the case for the composition range explored in the last part of the paper. Indeed, in some alloys, the role of screw dislocations may be predominant (see Ref. [26]). The authors should either show that the critical stress to move a screw dislocation is much lower than the edge dislocation for all the alloys or clearly state the limits of applicability of their method.

- l. 138-144: Considering Vegard's law for misfit volumes and the rule of mixture for the elastic constants seems like an oversimplifying assumptions that can significantly change the results of the model.

- l. 190: display problem with the references.

- l. 184-187: If Cr, Zr and Hf form intermetallics, why all the alloys discussed in this section contain large concentrations of these elements ?

- I feel like a conclusion is missing at the end of this manuscript to summarize the findings.

In brief, the results reported in this article are interesting but it seems that the authors do not show convincing enough evidence to demonstrate that the plasticity of RHEA are solely controlled by edge dislocations. Unless the authors can reply to all the above comments with convincing answers, I think they should bring more nuance to their claims and explain in details the limit of the theoretical approach used in the last part of the manuscript.

Reviewer #3 (Remarks to the Author):

The authors demonstrate that the high-temperature strength of some High-Entropy Alloys is controlled by the mobility of edge dislocations rather than screw dislocations. They then use a theory of HEA flow stress developed by some of the co-authors to perform a screening of alloy compositions in order to identify promising candidate alloys for high-strength high-temperature applications.

I think this is a very sound and well crafted study, with some minor issues which I am going to point out below. The applicability of the method is quite generic and the paper deserves some impact, in particular in view of the fact that there is some confusion in the literature regarding the mechanisms which control the high-temperature strength of compositionally complex bcc alloys. Given the unusual thoroughness of the presented study I am convinced that it can be very helpful to sort out this

confusion. I therefore think that NatComms is an adequate forum both in view of the visibility of the journal and in view of its open access nature.

Points of criticism:

1) The arguments which point out the important role of edge dislocations in plastic deformation of the investigated materials are convincing. But the state-of-the art in high temperature plasticity of bcc materials is not well represented, hence the authors somewhat overstate the novelty of this finding. In fact,

even though the literature abounds with statements of the type 'plasticity of bcc metals is controlled by screw dislocations' such statements are incorrect at temperatures above the so-called transition temperature, where the dislocation microstructure and hardening behavior even of pure bcc metals becomes conspicuously similar to the behavior of fcc metals, see e.g. Šesták, B., & Seeger, A. (1971). The Relationship between the Work-Hardening of BCC and FCC Metals. *physica status solidi (b)*, 43(1), 433-444). It has been equally long recognized that in the regime of elevated temperature, temperature effects iron out the mobility difference between edges and screws even in pure bcc metals, see e.g. Louchet, F., Kubin, L. P., & Vesely, D. (1979). In situ deformation of bcc crystals at low temperatures in a high-voltage electron microscope: Dislocation mechanisms and strain-rate equation, *Philosophical Magazine A*, 39(4), 433-454.

This is not to denigrate the finding that, in HEA, the same is also true, which in the present work is demonstrated by deploying an impressive array of methods such as to settle the question once and for all. Only, the result is not completely unexpected. In fact, as shown by some of the present authors in recent

work, the chemical disorder of HEAs most likely facilitates the motion of screws because of spontaneous formation of 'grown-in' kinks, whereas it impedes the motion of edges by solute effects. So what is true for the pure bcc metals is not quite unexpected for the bcc high-entropy alloys. The discussion should be re-written to put this point into perspective.

2) Some of the authors have, in recent years, proposed a theory of hardening in chemically complex alloys which may be fairly general and which constitutes a straightforward generalization of the classical solute hardening paradigm. They also proposed a generalization to temperature dependent behavior which accounts for the complex scaling behavior of collective pinning barriers. Owing to its semi-analytical nature and reliance on elemental properties, the theory is most suitable for rapid screening of a large spectrum of compositions, possibly even using brute-force approaches as in the present paper. The paper presents results of such an exercise, which in my opinion constitute an important progress.

However, here I have to criticize the singularly non didactical presentation of the results which borders on pure gibberish. Quoting from the text: "As shown in Figs. 4b and 4c, we find $\sim 6,000,000$ alloys with estimated strengths over 1 GPa at $T = 1,300$ K, and $\sim 1,300,000$ over 2 GPa, far exceeding the strengths of

any existing alloys". So I take a look at Figs 4b and 4c. These are two graphs which to the eye look very similar. The x axis has the label 'composition number' which runs from 0 to 10000 and which should be multiplied with 1000. As to the meaning of 'composition number' (which vaguely points at some labeling scheme) no information is given in the caption beyond the cryptic statement that 'The compositions indicate the average +/-

standard deviation for 1,000 compositions per bin' and the advertising blurb that 'the screening includes $>10,000,000$ compositions in the Nb-Mo-Ta-W-V-Cr-Ti-Zr-Hf-Al compositional space.' That may well be but WTF IS SHOWN IN THE FIGURE???

Looking at the y axes does not help much. On the top graphs of 4a and 4b I see an ascending strength and strength-to-weight, respectively. Left side = low = bad, right side = high = good. Below I see graphs for each element which presumably show

concentrations as they have no unit on the y axis, but there are multiple curves in each elemental graph and the meaning of these as well as the underlying ordering scheme is undecipherable.

I appreciate the difficulty encountered by the authors. Each alloy composition is a vector in a 10-dimensional space, and strength or STW ratio are functions on that space. Humans are not particularly good at visualizing functions on 10D-spaces. Yet the authors felt the need to present some kind of visual clue instead of just saying 'we screened everything and believe us, $\text{Mo}_5\text{W}_2.5\text{CrZrHf}$ and $\text{Mo}_2.5\text{W}_2.5\text{CrZrHf}$ are really really good'.

However, in doing so they caused me some mild headache when trying to comprehend the rationale of their plots and finally concluding that most likely there is none.

So please, make an attempt to better visualize this part of your work, or at least to explain the presented plots. In an age of data science tools like PCA are commonplace which hopefully might help to identify important and eliminate unimportant variables in high dimensional spaces. You have 10^7 data points, so now your task is to process and present them in a manner that a mortal like myself can make sense of them. Not having the data to play with, I can unfortunately only offer limited help. (Please put them into a public repository after publishing the paper so I can play with them later). But pleeeeeeease do not show me Figures 4b and 4c again in present form lest I go dizzy.

In conclusion, I strongly recommend this paper for publication in NatComm once the above points have been addressed.

REPLY TO REVIEWER'S COMMENTS

We greatly appreciate the editor's and reviewers' thoughtful comments on our paper entitled "Strength Can be Controlled by Edge Dislocations in Refractory High-Entropy Alloys". We agree with all points made in the comments and have very carefully revised the manuscript, as described below:

Reviewer #1 (Remarks to the Author):

Comment #1

This manuscript reports temperature dependence of yield strengths of BCC HEAs: NbTaTiV and CrMoNbV, and attributes the high strength and high strength retention to edge dislocations. Through a computer search a million new alloy compositions are identified "for future exploration".

- In BCC metals, screw dislocations have non-planar core structure and hence, exhibit lower mobility as compared to edge dislocation. However, this is primarily true at low homologous temperatures. The elevated temperature strengths ($> 0.5T_m$), where thermal activation can overcome the Peierls barrier, cannot be explained based on low mobility of screw dislocations at low homologous temperatures. Hence, the authors claim that strength retention at elevated temperatures can be attributed to edge dislocations appears somewhat confusing.

[Reply]

We very much appreciate the reviewer's comments on *elemental* BCC metals. The conventional understanding of BCC *alloys*, both via theory and experiments, has been that screw dislocations dominate strengthening at all temperatures (see e.g., Suzuki (1980) "Solid solution hardening in body-centered cubic alloys", Dislocations in Solids) [1]. In alloys above a few % of solutes, it has

been understood that double-kink nucleation is easy, and that screw motion is controlled by the lateral kink glide; this is the opposite of the situation in pure BCC metals. Furthermore, it is the formation of observable cross-kinks/jogs that is envisioned to provide the strengthening of screws at higher temperatures (see below). Thus, our claim that edge dislocations play a critical role in complex BCC *alloys*, such as HEAs, and at all temperatures, is quite different from all prior works in the literature. Flow controlled by edge dislocations then enables high strength retention at elevated temperatures where the screw cross-kink/jog strengthening can be defeated by vacancy-mediated mechanisms.

Several of the authors have now derived and validated theories for the strengthening mechanisms for both screw (Ref. 31, Maresca F, Curtin WA. Theory of screw dislocation strengthening in random BCC alloys from dilute to “High-Entropy” alloys. *Acta Materialia* 2020, 182: 144-162.) [2] and edge (Ref. 21, Maresca F, Curtin WA. Mechanistic origin of high strength in refractory BCC high entropy alloys up to 1900K. *Acta Materialia* 2020, 182: 235-249.) [3] dislocations in random/high entropy alloys. For screw dislocations, Reference 31 demonstrates that a long screw dislocation in a random alloy will form kinks spontaneously even at temperature $(T) = 0$ K. Moreover, because of the non-planar/compact core structure of the screw dislocations, these spontaneous kinks can form on any of the 3 possible $\{110\}$ glide planes of the BCC $\langle 111 \rangle$ -zone axis. This feature gives rise to cross-kinks/jogs that form strong pinning points, which can only be overcome by the formation of self-interstitials and vacancies, see Figure 4 and the related text in Ref. 31. The self-interstitial formation energies for BCC metals are enormous, 5 – 10 eV, giving rise to high-temperature strength retention but only up to temperatures at which vacancy migration becomes feasible and annihilates thus the cross-kink strengthening. Therefore, as seen experimentally, BCC HEAs observed to be controlled by screws do not retain strength above ~

$T_m/2$. In contrast, the refractory NbMoTaW(V) alloys investigated in Ref. 21 do not show such drop. Our theory for edge dislocations captures the strengthening versus temperature up to high-T, and no vacancy mechanism is yet proposed to defeat this strengthening mechanism. To address the reviewer's concern/confusing, we have revised/added the above discussion in the main text, as shown below.

References for the above reply

- [1] Suzuki H. Solid solution hardening in body-centred cubic alloys. *Dislocations in solids* **4**, 191-217 (1980).
- [2] Maresca F, Curtin WA. Theory of screw dislocation strengthening in random BCC alloys from dilute to "High-Entropy" alloys. *Acta materialia* **182**, 144-162 (2020).
- [3] Maresca F, Curtin WA. Mechanistic origin of high strength in refractory BCC high entropy alloys up to 1900K. *Acta materialia* **182**, 235-249 (2020).

Revised/added paragraphs in the Main text

Page 10, Lines 189 – 195 in the Main text

"Screw dislocations remain important in some BCC HEAs, but the vacancy-driven unpinning of the high-strength screw cross-kinks is expected to lead to a loss of strength at high T³¹. Moreover, accurate inputs for screw theories are difficult to obtain^{32,33}, and hence, a computationally-guided search for new high-performance alloys based on edge-dislocation strengthening is a *computable* mechanistic path for the design and discovery of further new alloys with high strengths and high-temperature strength retention, independent of whether or not screw dislocations provide strengthening."

Comment #2

- The data in Figure 1 show three stages: steep increase in strength with decreasing temperatures at lower temperatures, near plateau at intermediate temperatures and gradual decline at elevated temperatures. It is not clear that a single mechanism can explain the entire temperature range. The model predictions (Fig 4) show a poor match with the experimental data.

[Reply]

Here and elsewhere, we acknowledge that the edge model (nor any other existing model for screw dislocations) does not capture the observed plateau in strength at intermediate temperatures. However, it is important to note that the edge model is entirely parameter-free (depending only on material properties, such as elastic moduli, solute misfit volumes, and alloy concentration). The parameter-free predictions for CrMoNbV are thus in reasonable agreement with the experiments. The mismatch at RT may be in part related to observed microfractures, and the predictions in this regime also depend on a line tension that is not well-established. There is no data above the “plateau region”. The theory also predicts the experimental observation that this alloy is the strongest studied to date at $T = 1,000\text{ }^{\circ}\text{C}$.

Returning to the origins of the plateau regime, we are currently investigating a dynamic strain aging mechanism for edge dislocations as a possible explanation. This mechanism would exist only over a temperature window, beyond which the strength would then decrease again with increasing temperature, as governed by the present theory but with a constant upward shift, as seen experimentally in several cases. This topic is well beyond the scope of the present paper.

In NbTaTiV, the measured presence of higher concentrations of interstitials (O and N) increases the strength above that of the interstitial-free alloy; this strengthening is well-established in BCC

alloys, including HEAs. We have thus presented (i) an estimate of the additional strengthening due to O and N, (ii) other experiments on NbTaTiV that have lower strengths than our present experiments that we believe are due to lower interstitial levels, and (iii) show the established experimental increments of strengthening found in TiZrHfNb alloys. These factors rationalize the difference between theory and experiment. We cannot provide more quantitative predictions of the effects of interstitials at this time because the very strong interactions of interstitials in the core of the dislocation is not easily determined quantitatively. To address the reviewer’s concern, we have revised/highlighted the above discussion in the main text related to Fig. 4a.

Revised/added paragraphs in the Main text

Page 8, Line 158 – Page 9, Line 170 in the Main text

“The predicted strength versus temperature for NbTaTiV *with no interstitial content* at the experimental strain rate of $10^{-3}/s$ is shown in Fig. 4a along with the present experiments and literature data. The theory trend is good but much lower than the present experiments, although comparable to the literature data at $T = 300$ K. We attribute this difference to the 1.6 at% O + N impurities in our alloy because it is known that 2 at% O or N strengthens a BCC HEA by $\sim 400 - 500$ MPa at $T = 293$ K (Figure 4a)²⁹. Furthermore, we have computed the tetragonal misfit strains for O and N in Nb, Ta, and V, via density functional theory (DFT), and they are all large ($\varepsilon_{11} \sim 0.60$; $\varepsilon_{22} = \varepsilon_{33} \sim -0.1$), consistent with high strengthening. Semi-quantitatively, using a new Nb-O interatomic potential³⁰ to compute the interaction of O with an *edge* dislocation in Nb and assuming the same interaction energies for the NbTaVTi alloy, the theory can be extended to include the addition of 1.6 at% O interstitials and predicts an increase in strength of ~ 300 MPa at $T = 300$ K, reaching a reasonable agreement with the experiment (Figure 4a).”

Page 9, Lines 179 – 182 in the Main text

“The RT prediction may be higher due to the line tension coefficient²¹ or microfractures in the alloy. As discussed previously, this model also does not capture the plateau at 973 - 1,273 K that is observed in many BCC HEAs²¹ but has been shown to capture the further strength decreases found in other alloys at higher temperatures.”

Comment #3

- The TEM characterization (Fig 2 and 3) only shows room temperature deformed samples. Also, no information is provided on the core structure of the dislocations either by atomistic modeling or by HRTEM.

[Reply]

Considering the modified-Williamson-Hall plot of NbTaTiV and CrMoNbV (Fig 2d and 3d), it shows that the dislocation type is mainly edge dislocation at higher strains over the room temperature to 1,173 K range. Also, the RT-deformed samples present the dominancy of edge dislocations. Wang *et al.* published “Multiplicity of dislocation pathways in a refractory multiprincipal element alloy” in Science, and they explained the high T behavior only based on RT TEM [1]. We believe that our neutrons and TEM data is sufficient to make the point about edge dislocations.

Besides, at higher T, the screw dislocations had lesser effect on mechanical properties while the edge dislocation is increasingly competitive [2]. Therefore, if the edge dislocation is more important at RT like our samples, the effect of screw dislocation should attribute minor to the mechanical properties.

Regarding the core structure of the dislocations in HEAs, it is a very challenging work to do. Even in simple metals, the mechanical properties of Ti and Zr were explained based on the TEM image and simulation while the dislocation core structures were not shown [3]. The situation is the same in Wang et al.'s paper, which mainly used the TEM and simulation to characterize the dislocation type in the BCC HEA [1]. Especially, high-strength BCC HEAs like our samples exhibited large lattice distortion and were highly strained, which even make the dislocation core imaging even more difficult, and TEM has been found sufficient to support our claim that edge dislocations have high strength and feature prominently in the deformation.

References for the above reply

- [1] Wang F, Balbus G. H, Xu S, Su Y, Shin J, Rottmann P. F, Knipling K. E, Stinville, J. C, Mills L. H, Senkov O. N, Beyerlein I. J, Pollock T. M, and Gianola D.S. Multiplicity of dislocation pathways in a refractory multiprincipal element alloy. *Science* **370**, 95-101 (2020).
- [2] Maresca F, Curtin WA. Theory of screw dislocation strengthening in random BCC alloys from dilute to “High-Entropy” alloys. *Acta materialia* **182**, 144-162 (2020).
- [3] Clouet E, Caillard D, Chaari N, Onimus F, and Rodney D. Dislocation locking versus easy glide in titanium and zirconium. *Nature Materials* **14**, 931-936 (2015).

Comment #4

- In Fig. 1, comparison of yield strengths for different materials would be more meaningful if the strength values were normalized by modulus.

[Reply]

We are trying to make quantitative comparisons between theory and experiment, and to show actual strengths of different alloys. We believe normalizing by modulus would prevent such direct comparisons, especially since moduli are only not widely available and, if available, mainly at RT.

It is interesting to note that the screw theories for strengthening do not actually depend on modulus. Undoubtedly, the modulus plays some role within the parameters that enter screw theories, but it is not explicit. So, normalizing by modulus would probably be misleading in alloys that are controlled by screw dislocations. The edge theory depends directly on moduli, as seen in our reduced model of Eqs. 4 and 5. Hence, the dependence of strength on modulus is quite explicit here and normalizing by modulus does not convey additional understanding.

Comment #5

- References #16 and #25 appear to be the same.

[Reply]

We really appreciate the reviewer's pointing out the critical mistake. The Reference #25 has been removed.

Reviewer #2 (Remarks to the Author):

Comment #1

In this manuscript, the authors present experimental and theoretical arguments to show that the yield stress of refractory HEA is controlled by the glide of edge rather than screw dislocations. Based on this, the authors scan a large composition range of refractory HEA and propose new compositions with expected outstanding strength at high temperatures.

The arguments proposed by the authors show that edge dislocations are relevant and should not be discarded - in contrast with pure BCC metals where the kink-pair nucleation on screw dislocation is the plasticity-controlling process. Based on the points listed below, I will argue that the results presented in this manuscript are however not strong enough to discard the role of screw dislocations in these alloys, which appears as an implicit assumption to apply the edge-based theory to explore compositions as performed in the last part of the paper.

[Reply]

We thank the reviewer for the assessment of our work for this remark. We did not intend to give the impression that screw dislocations can always be discarded. In fact, we have developed theories of strengthening for both edge (Ref. 21, Maresca F, Curtin WA. Mechanistic origin of high strength in refractory BCC high entropy alloys up to 1900K. *Acta Materialia* 2020, 182: 235-249.) [1] and screw dislocations (Ref. 31, Maresca F, Curtin WA. Theory of screw dislocation strengthening in random BCC alloys from dilute to “High-Entropy” alloys. *Acta Materialia* 2020, 182: 144-162.) [2] in BCC random HEAs and dilute alloys and shown that some alloys are predicted to be screw-controlled and others edge controlled. The edge-controlled alloys are, in cases studied to date, able to retain strength up to very high temperature, whereas screw-controlled

alloys are expected to lose strength at high T due to a vacancy-mediated mechanism. Since it is widely believed that only screw dislocations are important in BCC alloys, the main point of this paper is to show, via experiments and theory, that edge dislocations *can* control the yield strength of BCC HEAs. A theory for the edge then shows reasonable quantitative agreement and enables search for promising compositions for high temperature strengths.

Let us step back and think broadly. There are three possibilities: (1) screws dominate the strength, (2) edges dominate the strength, or (3) screws and edges are comparable. Thus, a predictive theory for edge dislocations is a *lower bound* on the strength, independent of whether screws might provide comparable strengthening or even dominate the strengthening. If an edge theory, with no adjustable parameters, reasonably predicts the strength vs. temperature of an alloy, then it could not be the case that screw dislocations dominate (because the alloy strength would have to be higher, and it is not).

Now we then need to recognize that there is no parameter-free screw theory and no reduced model for screw strength; these are needed to execute any search over millions of alloys. So, the only path forward at present is to execute a search based on the edge theory, which provides a lower bound for some alloys. This search may exclude alloys that are strong due to screw strengthening, but we have absolutely no way to determine that this might be the case.

In the above framework, it can be seen that the singular focus of the literature on screw dislocation dominance in BCC alloys has probably been a hindrance to finding strong high-T alloys.

To address the reviewer's concern, we have corrected the manuscript throughout to avoid ambiguities.

References for the above reply

- [1] Maresca F, Curtin WA. Mechanistic origin of high strength in refractory BCC high entropy alloys up to 1900K. *Acta materialia* **182**, 235-249 (2020).
- [2] Maresca F, Curtin WA. Theory of screw dislocation strengthening in random BCC alloys from dilute to “High-Entropy” alloys. *Acta materialia* **182**, 144-162 (2020).

Revised/added paragraphs in the Main text

Page 4, Lines 50 – 54 in the Main text

“Note that screw dislocations can be important in some BCC HEAs¹⁵ but here we highlight that (i) the usually-overlooked edge dislocations can control the yield strength in some BCC HEAs, (ii) unlike screws, edge strengthening can convey high-T strength retention, and (iii) again unlike screws, an accurate parameter-free edge theory exists that enables alloy design.”

Comment #2

In addition, a recent publication (Ref. [15] of the present manuscript) shows the importance of dislocation glide on higher order glide systems. Even if this reference is cited in the manuscript, the results reported here are not discussed in regard of these findings. Please find below a tailed list of comments that need to be addressed:

- 1. 50-53: These statements need to be discussed in regards of the findings of Ref. [15].

[Reply]

We thank the reviewer for the assessment of our work for this remark. We have clarified that, indeed, we do not claim that edge dislocations control the strength of *all* BCC HEAs. But they can control the strength, which is important because their role has been typically overlooked.

Regarding Ref. 15 [Wang F, *et al.* Multiplicity of dislocation pathways in a refractory multiprincipal element alloy. *Science* **370**, 95-101 (2020).], our work was originally submitted for publication long before Ref. 15 was submitted. Thus, our work is completely independent of Ref. 15. Ref. 15 reports evidence of dislocation glide in alloys on {112} and {123} planes, which has long been widely reported to occur in elemental BCC metals (see e.g., Table I in the 1971 paper by Rosenberg and Piehler, <https://doi.org/10.1007/BF02662666>, as also shown below) [1].

Table I. Average Values of M for Various Slip Modes

Slip Mode	No. of Slip Planes per $\langle 111 \rangle$ Direction	M_{ave}	Ref.
{110} $\langle 111 \rangle$	3	3.067	6, 2
{112} $\langle 111 \rangle$	3	2.954	2
{123} $\langle 111 \rangle$	6	2.803	2
mixed {110, 112, and 123} $\langle 111 \rangle$	12	2.754	2
approximate pencil glide	20	2.748	11
pencil glide	∞	2.733	present work

Interestingly, Ref. 15 (published in October 2020) reports evidence of non-screw dislocations but, unlike the present paper, those authors were fully aware of, but made no comparison to, our theory of edge dislocations (that first appeared ~2 years earlier <https://arxiv.org/abs/1901.02100v1>) [2], which is to, our best knowledge, the only mechanistic model for strengthening of BCC HEAs that is based on edge dislocations. To address reviewer's suggestion in regards of the findings of Ref. [15], we have clarified the statements in light of the above discussion.

References for the above reply

- [1] Rosenberg J, Piehler H. Calculation of the Taylor factor and lattice rotations for bcc metals deforming by pencil glide. *Metallurgical Transactions* **2**, 257-259 (1971), <https://doi.org/10.1007/BF02662666>.

[2] Maresca F, Curtin WA. Mechanistic origin of high retained strength in refractory BCC high entropy alloys up to 1900K. *arXiv preprint arXiv:1901.02100*, (2019), <https://arxiv.org/abs/1901.02100v1>.

Revised/added paragraphs in the Main text

Page 2, Line 23 – Page 3, Line 30 in the Main text

“Recent TEM work has excitingly shown the MoNbTi alloy to deform by dislocations beyond the usually-assumed $(110)\langle 111 \rangle a/2$ screw dislocations^{15, 16} with few suggestions for alloy discovery. Here, we show that the high strength and high-T strength retention of both the recent NbTaTiV and new CrMoNbV HEAs (Figure 1) are controlled by edge dislocations. Our findings are unexpected because screw dislocations are widely understood to control plastic flows in BCC elemental metals and dilute alloys^{17, 18}, although there were hints in the literature that edge dislocation motion might be hindered by solutes in some low/moderate-concentration alloys¹⁹⁻²².”

Page 4, Lines 49 – 54 in the Main text

“We now demonstrate that edge dislocations, rather than screw dislocations, can control the plasticity in both NbTaTiV and CrMoNbV. Note that screw dislocations can be important in some BCC HEAs¹⁵ but here we highlight that (i) the usually-overlooked edge dislocations can control the yield strength in some BCC HEAs, (ii) unlike screws, edge strengthening can convey high-T strength retention, and (iii) again unlike screws, an accurate parameter-free edge theory exists that enables alloy design.”

Comment #3

- l. 54-83: I am not an expert of the CMWP method. How reliable is this technique and has it been applied successfully to other metal and alloys? Is there a reference where this method has been applied to BCC metals and BCC dilute alloys and demonstrate that the plasticity is controlled by screw dislocations? Indeed, the large error bars and the results obtained in the case of CrMoNbV (finding $q=-10$ instead of belonging to $[-2,2]$) appears as worrisome to a naive reader.

[Reply]

The CMWP procedure has been developed during the past 40 years based on the theoretical work of Krivoglaz [1] and Wilkens [2]. The method is modeling diffraction peak profiles, employing physically well-established profile functions. It has been widely used by a few hundred different authors to determine dislocation densities, slip mode and Burgers vector character in FCC, BCC, and HCP crystals [3]. Specifically, it was used to determine the dislocation structure in plastically-deformed BCC Nb and Ta [4]. The CMWP method strongly relies on the shape of the tail regions of a peak profile. In the case of *in-situ* experiments, there is always a tradeoff between sufficient counts depending on the data collection time and the accuracy of the straining procedure. The consequence of these experimental restrictions is that the counts in the tail regions of peak profiles may not be ideal. These experimental circumstances are the reason for the somewhat larger errors in the q -parameter values, which determine the fractions of dislocations with screw or edge character. However, the general conclusions about the overwhelming edge character are not influenced by these experimental circumstances. To give fundamental understanding of CMWP procedure for the readers, we have added the sentence and references, as shown below.

References for the above reply

- [1] M. A. Krivoglaz, in *X-ray and Neutron Diffraction in Nonideal Crystals*. Eds. Baryakhtar, V.G., Moss, S.C., Ivanov, M.A. & Peisl, J., Springer, Berlin, Heidelberg, (1996).
- [2] M. Wilkens, Theoretical aspects of kinematical X-ray diffraction profiles from crystals containing dislocation distributions, in *Fundamental Aspects of Dislocation Theory*, edited by J. A. Simmons, R. deWit & R. Bullough, Vol. II, National Bureau of Standards (US) Special Publication No. 317, pp. 1195-1221. Washington, DC: US Government Printing Office (1970).
- [3] G. Ribárik, B. Jóni, T. Ungár, The convolutional multiple whole profile (CMWP) fitting method, a global optimization procedure for microstructure determination, *Crystals*, **623** (2020) doi:10.3390/cryst10070623.
- [4] B. Jóni, E. Schafler, M. Zehetbauer, G. Tich, Ungár, Correlation between the microstructure studied by X-ray line profile analysis and the strength of high-pressure-torsion processed Nb and Ta, *Acta Mater.* **61** (2013) 632–642.

Added paragraphs in the Main text

Page 4, Lines 67 – 69 in the Main text

“Note that the CMWP procedure has been widely applied to determine dislocation densities, slip modes, and Burgers vector character in face-centered-cubic (FCC), BCC, and hexagonal-close-packed (HCP) crystals^{25, 26}.”

Page 5, Lines 85 – 88 in the Main text

“There are larger errors in the q-parameter values, which determine the fractions of dislocations with screw or edge character. However, the general conclusions about the overwhelming edge character are not influenced by the experimental uncertainty.”

Added References

25. Ribárik G, Jóni B, Ungár T. The convolutional multiple whole profile (CMWP) fitting method, a global optimization procedure for microstructure determination. *Crystals* **10**, 623 (2020).
26. Jóni B, Schafler E, Zehetbauer M, Tichy G, Ungár T. Correlation between the microstructure studied by X-ray line profile analysis and the strength of high-pressure-torsion processed Nb and Ta. *Acta materialia* **61**, 632-642 (2013).

Comment #4

- 1. 84-100: The STEM results presented here do not appear very conclusive. Looking at Fig. 2.e and 2.f, it seems that the microstructures of both alloys display long screw segments, at least in comparable proportion than edge dislocations. How is obtained the 75% and the 56% of edge component discussed in the manuscript ? Aren't these numbers weighted by dislocation length ?

[Reply]

In the manuscript, the dislocation type ratio was obtained by calculating the number of edge and screw dislocation types without weighting by its length. The process details were described, as below:

The TEM samples were thinned near the [110] direction with focused ion beam (FIB). In TEM, the samples were tilted for about 5° in the azimuthal angle according to [110] with -110 or 101 fixed to increase the angular difference in the dislocation type, as shown in Figs. 2.e and 2.f. On the stereographic projection, the pure edge, pure screw, and both possible dislocation-line directions were marked as red, blue, and green point. The black dislocation lines on the ABF-STEM images were marked, and their angular information was collected. The dislocation lines were sorted to the dislocation-line direction spots on the stereographic projection if the angular error was under 5 degrees, and their dislocation types were obtained. The ratio of edge dislocations was the number of edge dislocations divided by number of both edge and screw dislocations, and vice versa. The green case, which contained both possible dislocation lines, was excluded during the ratio calculation.

The ratio of dislocation types was not weighted to the dislocation line length because in the TEM measurement, the observed dislocation lines were from the projection of dislocation lines, and the length on the image was not the actual dislocation length. Therefore, the length-weighted dislocation type ratio may not represent the actual dislocation-type ratio. Rather, the number ratio should be closer to the actual dislocation type ratio. As suggested by the reviewer, we have added new statement to clearly indicate the measurement method of dislocation components.

Added paragraphs in the Supplementary Information

Page 16 in the Supplementary Information

“The ratio of dislocation types was not weighted to the dislocation-line length because in the TEM measurement, the observed dislocation lines were from the projection of dislocation lines, and the length on the image was not the actual dislocation length. Therefore, the length-weighted

dislocation type ratio may not represent the actual dislocation-type ratio. Rather, the number ratio should be closer to the actual dislocation-type ratio.”

Comment #5

- 1. 84-100: These results should be discussed in light of the findings of Ref. [15]. Did the author find evidence of other slip systems for these alloys ?

[Reply]

Based on the stereographic projection result (Figs. 2f and 3f), $\langle 111 \rangle a/2$ dislocation on $\{110\}$ and $\{112\}$ slip planes were frequently observed in NbTaTiV and CrMoNbV (Figs. 2e and 3e). We could not find the slip systems other than the systems mentioned in Ref. [15] [Wang F, *et al.* Multiplicity of dislocation pathways in a refractory multiprincipal element alloy. *Science* **370**, 95-101 (2020)]. To address reviewer’s suggestion, we have added new statement in the main text, as shown in below.

Added statements in the Main text

Page 6, Lines 103 – 105 in the Main text

“Note that $\langle 111 \rangle a/2$ dislocations on $\{110\}$ and $\{112\}$ slip planes were frequently observed in NbTaTiV and CrMoNbV, and we found the same type of slip systems with previously studied work by Wang *et al.*¹⁵.”

Comment #6

- Fig. 2.e: green segments appear in Fig. 2.e and it is not explained what they represent.

[Reply]

The green segments represented the dislocation lines that could be either edge or screw dislocation. For example, the $\langle 100 \rangle$ screw dislocation and $\langle 110 \rangle \{112\}$ edge dislocation may have the same projected dislocation line direction. Therefore, the green lines were not included in the ratio calculation. We have added the explanation of green segments in figure captions.

Added statements in the Main text

Page 21, Lines 440 – 441 and Page 24, Lines 459 – 460 in the Main text

“The green lines/symbols represented the dislocations that could be either edge or screw, which was excluded in the ratio calculation.”

Comment #7

- Fig. 2.d and 3.d: the label of the x-axes are inconsistent.

[Reply]

We really appreciate the reviewer’s kindly pointing out the label in Figs. 2d and 3d. We have modified Fig. 2d with the x-axis of $KC^{0.5}$, which becomes consistent with Figs. 2d and 3d.

Modified Figures 2 and 3 in the Main text

NbTaTiV High-entropy Alloy

Figure 2. Neutron diffraction and TEM experiments showing dominance of edge dislocations in NbTaTiV. **a**, Neutron-diffraction patterns presenting interplanar spacings with peaks indexed for the BCC structure of NbTaTiV, at temperatures of 293 K, 973 K, and 1,173 K. **b**, Axial and transverse lattice strains versus applied load, shown as applied stress versus lattice strain so that the slopes correspond to the planar Young's moduli (E_{hkl}) and Young's moduli/Poisson's ratios (E_{hkl}/ν_{hkl}), as indicated for the $\{110\}$, $\{200\}$, $\{211\}$, and $\{310\}$ planes, respectively, at $T = 293$ K. The onset of plastic yielding (yield stress) is presented as the dashed line. **c**, Evolution of q parameters as a function of plastic strain, which are obtained from the Convolutional Multiple Whole Profile (CMWP) fitting. Dashed lines indicate values of q parameters for edge and screw character dislocations, considering a 15 % error margin for the elastic constants. **d**, Modified-Williamson-Hall plot, FWHM versus $KC^{0.5}$ at plastic strains of 1.8 %, 6.8 %, and 11.8 %, respectively, at $T = 293$ K. The plots were obtained from the physical profiles calculated by the CMWP procedure, considering free of the instrumental effects on the FWHM data. The pattern of the undeformed specimen was applied to the CMWP procedure as an instrumental pattern. The much better agreement of the data with the edge analysis demonstrates the dominance of edge dislocations. **e**, Annular-Bright-field (ABF)-STEM image of NbTaTiV at a plastic strain of 11.8 % with a two-beam condition near $Z = [1\bar{1}3]$ and $\vec{g} = (110)$. All straight dislocation lines longer than 5 nm are highlighted by blue and red lines, corresponding to their identifications as edge and screw dislocations, respectively. **f**, Stereographic projection related to the $[110]$ orientation, where $[110]$ has been aligned with images in (e). All possible dislocations are indicated, and those corresponding to the images in (e) are highlighted in bold. The degrees indicate the angle with respect to the $[110]$ direction. Blue lines/blue symbols present those dislocations identified as edge,

and red lines/symbols indicate those dislocations identified as screw. The green lines/symbols represented the dislocations that could be either edge or screw, which was excluded in the ratio calculation. As a result, $\sim 75\%$ of the dislocations are identified as edge.

CrMoNbV High-entropy Alloy

Figure 3. Neutron diffraction and TEM experiments showing dominance of edge dislocations in CrMoNbV. **a**, Neutron-diffraction patterns presenting interplanar spacings with peaks indexed for the BCC structure of CrMoNbV, at temperatures of 293 K and 1,173 K. **b**, Axial and transverse lattice strains versus applied load, shown as applied stress versus lattice strain so that the slopes correspond to the planar Young's moduli (E_{hkl}) and Young's moduli/Poisson's ratios (E_{hkl}/ν_{hkl}), as indicated for the $\{110\}$, $\{200\}$, $\{211\}$, and $\{310\}$ planes, respectively, at $T = 293$ K. The onset of plastic yielding (yield stress) is presented as the dashed line. **c**, Evolution of q parameters as a function of plastic strain, which are obtained from the Convolutional Multiple Whole Profile (CMWP) fitting. Dashed lines indicate values of q parameters for edge and screw character dislocations, considering a 15 % error margin for the elastic constants. **d**, Modified-Williamson-Hall plot, FWHM versus $KC^{0.5}$ at plastic strains of 0.8 %, 4.2 %, and 5.3 %, respectively, at $T = 293$ K. The plots were obtained from the physical profiles calculated by the CMWP procedure, considering free of the instrumental effects on the FWHM data. The pattern of the undeformed specimen was applied to the CMWP procedure as an instrumental pattern. The much better agreement of the data with the edge analysis demonstrates the dominance of edge dislocations. **e**, Annular-Bright-field (ABF)-STEM image of CrMoNbV at a plastic strain of 4.2 % with a two-beam condition near $Z = [1\bar{1}3]$ and $\bar{g} = (110)$. All straight dislocation lines longer than 5 nm are highlighted by blue and red lines, corresponding to their identification as edge and screw dislocations, respectively. **f**, Stereographic projection related to the $[110]$ orientation, where $[110]$

has been aligned with images in (e). All possible dislocations are indicated, and those corresponding to the images in e are highlighted in bold. The degrees indicate the angle with respect to the [110] direction. Blue lines/blue symbols present those dislocations identified as edge, and red lines/symbols indicate those dislocations identified as screw. The green lines/symbols represented the dislocations that could be either edge or screw, which was excluded in the ratio calculation. As a result, ~ 55% of the dislocations are identified as edge.

Comment #8

- I. 110-158: the applicability of the edge model relies on the assumption that the role of the screw dislocations can be discarded. Because of the above comments, the experimental proof do not appear convincing enough to discard the role of screw dislocations in these alloys. Moreover, the findings of ref. [15] suggest the important role of other slip system. Could this model be applied to other slip system in order to clarify their roles?

[Reply]

We have addressed the role of screw dislocations in the previous Comment #1. The edge theory could indeed be applied to other slip systems, but that has not been done to date. In part, this is premature since it is first necessary to show the importance of the edge dislocations for the primary and widely accepted/observed slip system, as done here.

As mentioned in response to the reviewer's introductory comments, we have published two theory works treating the strengthening mechanisms of both edge (Ref. 21, Maresca F, Curtin WA. Mechanistic origin of high strength in refractory BCC high entropy alloys up to 1900K. *Acta Materialia* 2020, 182: 235-249.) [1] and screw dislocations (Ref. 31, Maresca F, Curtin WA. Theory of screw dislocation strengthening in random BCC alloys from dilute to "High-Entropy" alloys. *Acta Materialia* 2020, 182: 144-162.) [2] in BCC "random" HEAs. The analytical model in Ref. 31 does consider multiple slip mechanisms on {110} planes, leading to a microscale {110}

or $\{112\}$ slip (but higher index slip planes are not ruled out as a priority). Thus, we do not discard the role of screw dislocations. As we show in Ref. 31, because of their compact core and the random alloy, the screw dislocations are spontaneously kinked (see also Graphical Abstract of Ref. 31), and they can kink on multiple $\{110\}$ planes because of the symmetry of their core. Although not studied in the detail in Ref. 31, we expect that glide of screw dislocations in random alloys can lead to glide on high index planes as a sum of elementary $\{110\}$ steps. However, this is not the focus of this paper, which rather discusses the possible role of edge dislocations.

References for the above reply

- [1] Maresca F, Curtin WA. Mechanistic origin of high strength in refractory BCC high entropy alloys up to 1900K. *Acta materialia* **182**, 235-249 (2020).
- [2] Maresca F, Curtin WA. Theory of screw dislocation strengthening in random BCC alloys from dilute to “High-Entropy” alloys. *Acta materialia* **182**, 144-162 (2020).

Comment #9

- Fig. 4: The comparison of the red theoretical curve with experimental points for NbTaTiV accounting for interstitials appears reasonable. However, the comparison of experimental points with the theoretical curve for the CrMoNbV alloy appears less convincing. Indeed, while the theory gives satisfactory results for intermediate temperatures, it seems that the global temperature dependence of the yield stress is not reproduced by the model: the model overestimates (underestimates) the yield stress at low (high) T respectively. This tendency is not discussed in the manuscript and might suggest that the model does not include all the possible mechanisms (as the role screw dislocation and other slip systems) that may play an important role at low or high temperature.

[Reply]

Here and elsewhere, we acknowledge that the edge model (nor any other existing model for screw dislocations) does not capture the observed plateau in strength at intermediate temperatures. However, it is important to note that the edge model is entirely parameter-free (depending only on material properties, such as elastic moduli, solute misfit volumes, and alloy concentrations). The parameter-free predictions for CrMoNbV are thus in reasonable agreement with the experiments. The mismatch at \sim RT may be in part related to observed microfractures, and the predictions in this regime depend on a line tension that is not well-established. There is no data above the “plateau region”. The theory also predicts the experimental observation that this alloy is the strongest studied to date at $T = 1,000$ °C.

Returning to the origins of the plateau regime, we are currently investigating a dynamic strain aging mechanism for edge dislocations as a possible explanation. This mechanism would exist only over a temperature window, beyond which the strength would then decrease again with increasing temperature, as governed by the present theory but with a constant upward shift, as seen experimentally in several cases. This topic is well beyond the scope of the present paper.

To address the reviewer’s concern, we have revised/highlighted the above discussion in the main text related to Fig. 4a.

Revised/added paragraphs in the Main text

Page 9, Lines 179 – 182 in the Main text

“The RT prediction may be higher due to the line tension coefficient²¹ or microfractures in the alloy. As discussed previously, this model also does not capture the plateau at 973 - 1,273 K that

is observed in many BCC HEAs²¹ but has been shown to capture the further strength decreases found in other alloys at higher temperatures.”

Comment #10

- Even if we concede that the plasticity of CrMoNvV and NbTaTiV are controlled by edge dislocations, it may not be the case for the composition range explored in the last part of the paper. Indeed, in some alloys, the role of screw dislocations may be predominant (see Ref. [26]). The authors should either show that the critical stress to move a screw dislocation is much lower than the edge dislocation for all the alloys or clearly state the limits of applicability of their method.

[Reply]

As commented earlier, there are three possibilities: screws dominate the strength, edges dominate the strength, or screws and edges are comparable. Thus, a predictive theory for edge dislocations is a *lower bound* on the strength, independent of whether screws might provide comparable strengthening or even dominate the strengthening. If an edge theory, with no adjustable parameters, reasonably predicts the strength vs. temperature of an alloy, then it could not be the case that screw dislocations dominate (because the alloy strength would have to be higher, and it is not). Since there is no parameter-free screw theory and no reduced model for screw strength, the only path forward is to execute a search based on the edge theory, which provides a lower bound for some alloys. This search may exclude alloys that are strong due to screw strengthening, but we have absolutely no way to determine that this might be the case.

Comment #11

- 1. 138-144: Considering Vegard's law for misfit volumes and the rule of mixture for the elastic constants seems like an oversimplifying assumptions that can significantly change the results of the model.

[Reply]

We have investigated these assumptions in some detail in Ref. 21 (Maresca F, Curtin WA. Mechanistic origin of high strength in refractory BCC high entropy alloys up to 1900K. *Acta Materialia* 2020, 182: 235-249.) [1], see Table 1 below and the related text, and Vegard's Law in the BCC HEAs has been reasonably validated by a number of other workers.

Table 1

Solute misfit volumes and elastic constants for the alloys studied, as computed using (i) Density Functional Theory (data kindly provided by Dr. B. Yin [35], using methods detailed in Ref. [36]), (ii) Vegard's law (misfits) or a rule-of-mixtures ROM (elastic constants), (iii) the true random alloy as described by EAM potentials, and (iv) the average-alloy EAM potential.

Mo-Nb-Ta-V-W	Method	a_{bcc}	ΔV_{Mo}	ΔV_{Nb}	ΔV_{Ta}	ΔV_V	ΔV_W	C_{11}	C_{12}	C_{44}
20-20-20-20-20	DFT	3.192	-0.824	1.848	1.882	-2.380	-0.526	338	164	51
	Vegard/ROM	3.192	-0.628	1.713	1.877	-2.484	-0.478	346.8	157.7	90.5
	EAM, Random	3.201	-0.956	1.246	1.571	-1.547	-0.333	306.3	156.8	79.1
	EAM	3.2	-0.924	1.246	1.566	-1.495	-0.266	317.9	158.8	83
25-25-25-0.0-25	DFT	3.237	-1.251	1.153	1.132	-	-1.034	374	163	64
	Vegard/ROM	3.228	-1.293	1.135	1.168	-	-1.010	375.5	167.3	101.6
	EAM, Random	3.223	-1.263	1.014	1.162	-	-0.914	350.6	168.9	93.2
	EAM	3.221	-1.218	1.019	1.181	-	-0.845	352.1	175.2	96.0
21.7-20.6-15.6-21-21.1 (Nominal Mo-Nb-Ta-V-W alloy)	Vegard/ROM	3.185	-0.628	1.8	1.833	-2.132	-0.348	355.6	156.7	92.4
	EAM, Random	3.195	-0.826	1.321	1.67	-1.484	-0.180	312.8	157.2	78.8
	EAM	3.194	-0.803	1.316	1.653	-1.434	-0.127	317.9	158.8	83
25.6-22.7-24.4-0.0-27.3 (Nominal Mo-Nb-Ta-V-W alloy)	Vegard/ROM	3.224	-1.229	1.199	1.232	-	-0.946	385.1	167.1	106
	EAM, Random	3.219	-1.175	1.056	1.197	-	-0.846	357.5	170.6	96.2
	EAM	3.217	-1.128	1.066	1.230	-	-0.774	358.8	174.4	97.7
24.9-25.8-26.6-22.7-0.0 (Nominal Mo-Nb-Ta-V alloy)	Vegard/ROM	3.205	-0.94	1.489	1.521	-2.444	-	300.8	146.6	72.8
	EAM, Random	3.211	-1.194	1.205	1.615	-1.962	-	264.7	144.4	66.9
	EAM	3.21	-1.156	1.215	1.627	-1.886	-	265.1	146.1	70.6
0.0-28.5-29.65-20.67-21.18 (Nominal Nb-Ta-V-W alloy)	Vegard/ROM	3.22	-	1.258	1.29	-2.675	-1.171	310.3	152.5	78.2
	EAM, Random	3.231	-	0.946	1.218	-2.046	-0.969	268.5	150	71.5
	EAM	3.23	-	0.95	1.205	-1.95	-0.938	267.7	149.3	75.6

We need to keep in mind that without simplified models, searching becomes nearly impossible.

There is nowhere near enough experimental data in this class of materials to execute any kind of “machine learning” approaches to obtaining the necessary quantities. Also, first-principles DFT

(putting aside the computational intensity) is not quite accurate enough for atomic or misfit volumes and is not at all accurate for C44 in BCC metals (this is well-known).

Comment #12

- l. 190: display problem with the references.

[Reply]

We really appreciate the reviewer's pointing out the critical mistake. It has been fixed, as shown below.

Revised/added paragraphs in the Main text

Page 11, Lines 209 – 212 in the Main text

“Future combinations of our design strategy with detailed thermodynamics^{35, 36} and added constraints (e.g., high ductility criterion) may lead to the discovery of new alloys that can achieve the multi-objective performance required in many critical engineering applications.”

Comment #13

- l. 184-187: If Cr, Zr and Hf form intermetallics, why all the alloys discussed in this section contain large concentrations of these elements ?

[Reply]

Indeed, as we have analyzed in the Supplementary Note 7, Cr, Zr, and Hf in considerable (equimolar) concentrations can lead to intermetallics formation. However, this is not the case for

the alloys mentioned in the manuscript, where the Cr, Zr, and Hf concentrations are much lower than Mo and W, which instead do not show the same tendency to form intermetallics. Since Cr, Zr, and Hf can contribute to large misfit volumes and hence, large strengthening, it is interesting to explore compositions that contain some of these elements.

We have clarified the conclusions of the manuscript and why non-equimolar concentration with some Cr, Zr, and Hf have been also considered, as shown below.

Clarified conclusion in the Supplementary Information

Pages 26 – 27 in the Supplementary Information

“For the equiatomic CrMoWZr, we predict phase separation at low temperatures into a mixture of BCC-Cr, Cr₂Zr.cF24, W₂Zr.cF24, and BCC-Mo. The transition to a single-phase BCC structure occurs at $T_0 = 2,300$ K. The melting temperature is not precisely known. We estimate it as 2,300 K by averaging the melting temperatures of the six equiatomic binaries. Thus, we predict the equiatomic BCC phase to be unstable at all temperatures below melting. Because Laves phases are major competitors to the HEA, we estimated the free energy of the cF24 Laves phase assuming a concentration, Mo₂Zr₆, on site 8a and Cr₆Mo₄W₆, on site 16d. This structure lies 82 meV/atom above the convex hull, suggesting that it could be stabilized by the entropy of mixing on the sublattices above $T_0 = 1,300$ K.

Cr and Zr are especially prone to the Laves-phase formation. Hence, we investigated the effect of moving off-stoichiometry on CrMo₂W₂Zr within a 24-atom supercell. Because the composition moves away from the Cr₂Zr and W₂Zr Laves phases, the distribution of ΔE_k values shifts downward by approximately 100 meV/atom, and we predict the formation of a single-phase BCC structure at $T_0 = 1,600$ K. At the same time, the melting temperature should rise because the composition is enriched in elements, Mo and W, whose melting temperatures are high. Thus, we

obtain a thermally-stable non-stoichiometric HEA over a wide temperature range. However, this analysis considers only a subset of potentially-competing phases that omit binary and ternary solid solutions. Hence, further investigation will be required to validate this prediction.

The situation is similar for CrHfMoW, whose phase separates at low temperatures into a mixture of BCC phases plus the Laves phase, HfW₂. The transition to a single BCC phase occurs at $T_0 = 2,400$ K, compared with our estimated melting temperature of 2,100 K. The high entropy Laves phase is stabilized above $T_0 = 1,350$ K. At the composition, CrHfMo₂W₂, the predicted T_0 drops to 1,200 K, while the melting temperature rises.”

Comment #14

- I feel like a conclusion is missing at the end of this manuscript to summarize the findings.

[Reply]

We thank the reviewer for this remark, and we have added the final part of the manuscript accordingly.

Revised/added paragraphs in the Main text

Page 11, Line 213 – 218 in the Main text

“To conclude, the present work, motivated by recent theory, demonstrates by means of an ample array of experimental techniques that edge dislocations can control the strength of BCC high-entropy alloys. This finding supports using a mechanistic theory based on edge-dislocation strengthening in BCC alloys to search for new, strong alloys. As shown in the current work, this

theory is a *validated* and *viable* way to perform the combinatorial search in the immense HEAs compositional space.”

Comment #15

In brief, the results reported in this article are interesting but it seems that the authors do not show convincing enough evidence to demonstrate that the plasticity of RHEA are solely controlled by edge dislocations. Unless the authors can reply to all the above comments with convincing answers, I think they should bring more nuance to their claims and explain in details the limit of the theoretical approach used in the last part of the manuscript.

We believe that we have addressed the main points raised by the Reviewer, including the role of screw dislocations and multiple slip planes, in the rebuttal letter and the revised manuscript. The limits of current theoretical approaches have been thoroughly discussed in the main text.

We really appreciate reviewer’s all points made in the comments. Due to reviewer’s critical comments and suggestions, the revised manuscript is now strongly developed by revisions.

Reviewer #3 (Remarks to the Author):

Overall Comment

The authors demonstrate that the high-temperature strength of some High-Entropy Alloys is controlled by the mobility of edge dislocations rather than screw dislocations. They then use a theory of HEA flow stress developed by some of the co-authors to perform a screening of alloy compositions in order to identify promising candidate alloys for high-strength high-temperature applications.

I think this is a very sound and well crafted study, with some minor issues which I am going to point out below. The applicability of the method is quite generic and the paper deserves some impact, in particular in view of the fact that there is some confusion in the literature regarding the mechanisms which control the high-temperature strength of compositionally complex bcc alloys. Given the unusual thoroughness of the presented study I am convinced that it can be very helpful to sort out this confusion. I therefore think that NatComms is an adequate forum both in view of the visibility of the journal and in view of its open access nature.

[Reply]

We really thank the Reviewer for the positive feedback.

Comment #1

The arguments which point out the important role of edge dislocations in plastic deformation of the investigated materials are convincing. But the state-of-the art in high temperature plasticity of bcc materials is not well represented, hence the authors somewhat overstate the novelty of this finding. In fact, even though the literature abounds with statements of the type 'plasticity of bcc metals is controlled by screw dislocations' such statements are incorrect at temperatures above the

so-called transition temperature, where the dislocation microstructure and hardening behavior even of pure bcc metals becomes conspicuously similar to the behavior of fcc metals, see e.g. Šesták, B., & Seeger, A. (1971). The Relationship between the Work-Hardening of BCC and FCC Metals. *physica status solidi (b)*, 43(1), 433-444). It has been equally long recognized that in the regime of elevated temperature, temperature effects iron out the mobility difference between edges and screws even in pure bcc metals, see e.g. Louchet, F., Kubin, L. P., & Vesely, D. (1979). In situ deformation of bcc crystals at low temperatures in a high-voltage electron microscope: Dislocation mechanisms and strain-rate equation, *Philosophical Magazine A*, 39(4), 433-454. This is not to denigrate the finding that, in HEA, the same is also true, which in the present work is demonstrated by deploying an impressive array of methods such as to settle the question once and for all. Only, the result is not completely unexpected. In fact, as shown by some of the present authors in recent work, the chemical disorder of HEAs most likely facilitates the motion of screws because of spontaneous formation of 'grown-in' kinks, whereas it impedes the motion of edges by solute effects. So what is true for the pure bcc metals is not quite unexpected for the bcc high-entropy alloys. The discussion should be re-written to put this point into perspective.

[Reply]

We agree with the analysis of the reviewer, and we have adjusted the manuscript accordingly. Indeed, in contrast with pure BCC metals, in BCC high-entropy alloys kinks are already present in the screw dislocation (Ref. 31, Maresca F, Curtin WA. Theory of screw dislocation strengthening in random BCC alloys from dilute to “High-Entropy” alloys. *Acta Materialia* 2020, 182: 144-162.) [1] and hence, there is no kink-pair nucleation mechanism that controls the strength. As for the references suggested by the reviewer, please note that our theory predicts the initial yield strength of BCC HEAs, thus not addressing work hardening. We have however referred to

early work by Statham, Koss, and Christian on $\text{Nb}_{(1-x)}\text{Re}_x$ and $\text{Nb}_{(1-x)}\text{Mo}_x$ and more recent work by Caillard on $\text{Fe}_{(1-x)}\text{Si}_x$ pointing to decreased edge dislocation mobility as a function of the increasing solute content.

In order to address reviewer's comments, we have referenced to earlier work pointing to the increased role of edge dislocations with alloying, as shown below.

References for the above reply

- [1] Maresca F, Curtin WA. Theory of screw dislocation strengthening in random BCC alloys from dilute to “High-Entropy” alloys. *Acta materialia* **182**, 144-162 (2020).

Revised/added paragraphs in the Main text

Page 3, Lines 28 – 30 in the Main text

“although there were hints in the literature that edge dislocation motion might be hindered by solutes in some low/moderate-concentration alloys¹⁹⁻²².”

Added references

19. Stephens JR. Dislocation structures in single-crystal tungsten and tungsten alloys. *Metallurgical and Materials Transactions B* **1**, 1293-1301 (1970).
20. Shields J, Gibala R, Mitchell T. Dislocation substructure in Ta-Re-N alloys deformed at 77 K. *Metallurgical Transactions A* **7**, 1111-1121 (1976).
21. Statham C, Koss D, Christian J. The thermally activated deformation of niobium-molybdenum and niobium-rhenium alloy single crystals. *Philosophical Magazine* **26**, 1089-1103 (1972).
22. Caillard D. A TEM in situ study of alloying effects in iron. II—Solid solution hardening caused by high concentrations of Si and Cr. *Acta materialia* **61**, 2808-2827 (2013).

Comment #2

Some of the authors have, in recent years, proposed a theory of hardening in chemically complex alloys which may be fairly general and which constitutes a straightforward generalization of the classical solute hardening paradigm. They also proposed a generalization to temperature dependent behavior which accounts for the complex scaling behavior of collective pinning barriers. Owing to its semi-analytical nature and reliance on elemental properties, the theory is most suitable for rapid screening of a large spectrum of compositions, possibly even using brute-force approaches as in the present paper. The paper presents results of such an exercise, which in my opinion constitute an important progress.

However, here I have to criticize the singularly non didactical presentation of the results which borders on pure gibberish. Quoting from the text: "As shown in Figs. 4b and 4c, we find ~ 6,000,000 alloys with estimated strengths over 1 GPa at $T = 1,300$ K, and ~ 1,300,000 over 2 GPa, far exceeding the strengths of any existing alloys". So I take a look at Figs 4b and 4c. These are two graphs which to the eye look very similar. The x axis has the label 'composition number' which runs from 0 to 10000 and which should be multiplied with 1000. As to the meaning of 'composition number' (which vaguely points at some labeling scheme) no information is given in the caption beyond the cryptic statement that 'The compositions indicate the average +/- standard deviation for 1,000 compositions per bin' and the advertising blurb that 'the screening includes >10,000,000 compositions in the Nb-Mo-Ta-W-V-Cr-Ti-Zr-Hf-Al compositional space.' That may well be but WTF IS SHOWN IN THE FIGURE???

Looking at the y axes does not help much. On the top graphs of 4a and 4b I see an ascending strength and strength-to-weight, respectively. Left side = low = bad, right side = high = good. Below I see graphs for each element which presumably show concentrations as they have no unit

on the y axis, but there are multiple curves in each elemental graph and the meaning of these as well as the underlying ordering scheme is undecipherable.

[Reply]

We thank the reviewer for the positive feedback.

We can agree that the figure is challenging, and that we did not allocate sufficient space to describing it in sufficient detail. In our defense, the reviewer seems to appreciate the difficulty of attempting to show results for 10,000,000 alloys in a compact form.

Therefore, we have modified Figure 4 in the main text, to better clarify the meaning of the axes and the labels, and include a zoom in to show 100 concentrations extracted from the screening. We have also extended the explanation in the caption and in Supp. Info. 6. We have moved the strength/density screening to Supp. Info. 6 to enable better clarity of Fig. 4. We hope that this revision and the responses below help better the understanding.

Here is the explanation of the figure in some detail. After computing the strength of 10,003,049 alloys, we have assigned a number to the compositions, from the lowest strength to the highest strength alloys. Thus, number 1 is the composition with the lowest strength, and number 10,003,049 with the highest strength. We then show yield strength *vs* composition number as the first (top) curve in Fig. 4b. The curve is naturally smooth because we have numbered the alloys from the lowest to the highest strength. About the same strength can be achieved by very different compositions (now shown in the revised Fig. 4c). The further curves in Figure 4b then refer to alloy compositions. The shaded area shows the range of elemental concentrations among 1,000 alloys that have, essentially, the same strength, with the range indicated as a mean and \pm one standard deviation. These curves show, for example, that most low-strength alloys do not contain

Cr (because Cr often has a large misfit volume and a high stiffness). In contrast, Ti shows an opposite trend: many low-strength alloys contain Ti while the super-high strength alloys contain minimal amounts of Ti. Nb and Al are also show similar trends with Ti.

In the revised figure, we have tried to clarify the meaning of the element concentration plots.

Revised Figure and Figure Caption

Figure 4 in Main text

Figure 4. Theory predictions of yield strengths in BCC HEAs. **a**, Yield strength vs. temperature, experiments and theory for NbTaTiV and CrMoNbV. **NbTaTiV**: Experiments on the NbTaTiV alloy with 1.15% O and 0.45% N: red triangles at a strain rate of 10^{-3} s^{-1} ; theory for an *interstitial-free* alloy: red line. Red square reported in Ref. 9 at $5 \times 10^{-4} \text{ s}^{-1}$ with no O or N impurity content reported. Black circles: experiments on TiHfZrNb at $T = 300 \text{ K}$ with and without 2% O and N that show strength increase up to $\sim 500 \text{ MPa}$ ²¹, comparable to the difference here between model predictions for the interstitial-free NbTaTiV and the experiments with 1.6% impurities. **CrMoNbV**: Experiments: orange triangles; theory: orange line; no impurities are detected in this alloy. The

strength of the new CrMoNbV at 1,173 K exceeds those of all previous reported alloys. **b** Theory predictions for $T = 1,300$ K strength vs. composition. As detailed in the Supplementary Note 6, computations of the yield strength have been performed for $> 10,000,000$ alloys. The alloys have been assigned with an increasing number as a function of the increasing strength. Thus, the lowest strength alloy is the number 1, and the highest strength alloy is the number 10,003,049. For better visualization, we have grouped the alloys into bins containing 1,000 compositions each (see the Supplementary Note 6). The lowest compositions are for alloys 1 to 1,000 and enter the 1st bin. The second bin contains alloys 1,001 to 2,000, etc. Within each bin, the average elemental content (Nb, Mo, Ta, and etc.) \pm standard deviation (according to a Gaussian distribution) is computed. In this panel, the shaded areas indicate the average \pm standard deviation of the composition, evaluated over the compositional bins. Thus, the y-axis is the elemental concentration. The screening includes $> 10,000,000$ compositions in the Nb-Mo-Ta-W-V-Cr-Ti-Zr-Hf-Al compositional space. **c** A zoom-in of the screening for 100 compositions around the 1,000,000 strongest alloys (between composition numbers of 9,005,000 and 9,005,100). Here, the actual alloy contents per composition number are shown. There are plenty of possible alloys that can have the same yield strength, but different compositions.

Revised/added paragraphs and figure in the Supplementary Information

Pages 20 – 21 in the Supplementary Information

“Note 6. Finding high-temperature strengths in the whole Cr-Mo-Nb-Ta-V-W-Ti-Zr-Hf-Al composition space.

“Figures 4b in the main text, as well as Figure S7, shows the predictions of the reduced theory (see the main text) as a function of the composition for $> 10,000,000$ compositions in the whole Cr-Mo-Nb-Ta-V-W-Ti-Zr-Hf-Al space. The analytical theory (Eqs. 2 and 3 with Eqs. 4 and

5) has been used to compute yield strength of over 10 million compositions in the whole Nb-Mo-Ta-W-V-Cr-Ti-Zr-Hf-Al family, at $T = 1,300$ K. Thus, the outcome of the calculations is a large table with >10 M entries (to be precise, 10,003,049 compositions). Each entry records the composition and the yield strength (Figs. 4b and S6a) or the ratio between the yield strength and density (Fig. S6b). A label has been assigned to the compositions, from the lowest to the highest strength alloys. Thus, the number 1 is the composition with the lowest strength, and the highest strength is the number 10,003,049. By plotting the yield strength vs. composition, the top curve in Fig. S6a is obtained. The curve is naturally “smooth” because the alloys have been numbered from the lowest to the highest strength (or strength/weight, Fig. S6b).

By following the alloy ranking from the lowest to the highest strength (or strength/ratio in Fig. S6b), it is known that on top of the yield strength, what is the alloy composition. By plotting directly, the Nb, Mo, etc. contents as a function of the alloy label (weakest to strongest), the composition vs. alloy number is not a smooth function, but is rather noisy, as shown in the zoom-in of Fig. S6a. This is because one can find alloys that have similar yield strengths, but very different compositions. Therefore, for the sake of the visualization, the elemental content vs. compositions plot has been “smoothened”, by following this operation: take the first 1,000 lowest strength compositions. Compute the average Nb, Mo, ... concentrations among these first 1,000 compositions, and store the result. Take the next 1,000 compositions, compute the average, and store the result. Proceed until one covers all the alloy compositions. Thus, as a final result, one obtains the average content of Nb, Mo, etc... per “bins” of 1,000 compositions. Hence, the bin-averaged composition can be plotted as a function of the bin number, which goes from 1 to $> 10,000$. During this averaging operation, the “noise” is lost. In order to assess how much an element can be varied within a bin, to obtain a similar yield strength, the standard deviation of the

yield strength has been computed per bin (by following a Gaussian distribution). The plot reports the average +/- the standard deviation, per bin, of the concentration of each element. Therefore, the y axis of the composition plots for Nb, Mo, Ta, etc. is dimensionless, because it corresponds to the elemental concentration. The screening is performed, using as the input of the single-elemental atomic volumes, elastic constants, and densities, all listed in Table S2. The alloy values are then computed, employing the rule of mixtures of the elemental values (see the main text).

For the case of Al, the atomic volume is $14,075 \text{ \AA}^3$, based on the work by Chen et al.⁶. The atomic volumes of all other elements are the same, as reported in the Ref.⁷. For the Ti and Zr, the values are obtained by extrapolating high-temperature (high-T) measurements to room temperature (RT), while Hf is obtained, employing the Vegard's law on Hf-HEAs. The atomic volumes adopted for Ti, Zr, and Hf are similar to those estimated in Ref.⁸, which were instead obtained by extrapolating the elemental values from binary alloys in the literature.

The cubic elasticity constants of the BCC Al are assumed to be equal to the FCC values. The Ti, Zr, and Hf values are taken from high-T phonon measurements (at 1,293 K, 1,188 K, and 2,073 K, respectively), see in Ref.⁹⁻¹¹.”

Figure S6. a. Theory predictions for $T = 1,300$ K strength vs. composition. As detailed in the Supplementary Note 6, computations of the yield strength have been performed for $> 10,000,000$ alloys. The alloys have been assigned with an increasing number as a function of the increasing strength. Thus, the lowest strength alloy is the number 1, and the highest strength alloy is the number 10,003,049. For better visualization, we have grouped the alloys into bins containing 1,000 compositions each. Here, as zoom-in is shown over one bin, which includes the compositions between 9,005,000 and 9,006,000. In the left panel, the compositions indicate the average \pm standard deviation for 1,000 compositions per bin. Thus, the non-dimensional y-axis is the elemental concentration. The zoom-in shows the raw data, which are characterized by large fluctuations in the composition for approximately the same strength. In the example shown, there are 1,000 compositions that can attain ~ 2.13 GPa at $T = 1,300$ K, and all elements of the Nb-Mo-Ta-W-V-Cr-Ti-Zr-Hf-Al compositional space can be used. The elements that are typically in the largest concentrations are Cr, W, and Mo, followed by Zr and Hf. **b.** Theory predictions for $T =$

1,300 K strength/density vs. composition. The plot is constructed the same as the panel **a**, by ranking the composition from the lowest to the largest strength/density ratio.

Comment #3

I appreciate the difficulty encountered by the authors. Each alloy composition is a vector in a 10-dimensional space, and strength or STW ratio are functions on that space. Humans are not particularly good at visualizing functions on 10D-spaces. Yet the authors felt the need to present some kind of visual clue instead of just saying 'we screened everything and believe us, Mo₅W_{2.5}CrZrHf and Mo_{2.5}W_{2.5}CrZrHf are really really good'. However, in doing so they caused me some mild headache when trying to comprehend the rationale of their plots and finally concluding that most likely there is none.

[Reply]

We appreciate the recognition that it is difficult to show 10^7 results. We do not expect readers to use precisely this graph, but rather to understand that there are indeed millions of super-strong alloys out there. We apologize for causing any headaches; indeed trying to present these results was a challenging task. We hope that the new explanatory notes and the new figure help the reader.

Comment #4

So please, make an attempt to better visualize this part of your work, or at least to explain the presented plots. In an age of data science tools like PCA are commonplace which hopefully might help to identify important and eliminate unimportant variables in high dimensional spaces. You

have 10^7 data points, so now your task is to process and present them in a manner that a mortal like myself can make sense of them. Not having the data to play with, I can unfortunately only offer limited help. (Please put them into a public repository after publishing the paper so I can play with them later). But pleeeeeeease do not show me Figures 4b and 4c again in present form lest I go dizzy.

[Reply]

Since there are many “very strong” compositions that contain most of the elements (see the new Figure 4c), and since composition is the only variable, we do not see any path for dimensional reduction. Some elements are less important for high strength at high T (Nb, Ti, Al, not surprisingly) but they may be desirable for other properties (ductility, oxidation resistance, density, and etc.). Hence, it is important to show that they can be incorporated to some degree while maintaining high-T strength.

The data in Figure 4 is obtained, using simple analytic formulations (Eqs. 2-5) and inputs, and thus, should be easily implemented with a short MATLAB code. The input to construct the database has now been added to the Supplementary Information to enable readers to explore. By using our MATLAB code, a user could implement further desired constraints or limit the study to any subset of elements, etc. very easily.

The $> 10,000,000$ records will be made available upon publication on a public repository (the file size is too large to include them into Supplementary Information) such that readers do not even need to use the MATLAB code.

We have created the public repository with the raw data and a MATLAB code to predict the yield strengths of BCC HEAs as a function of temperature, and we have included links to them to the Supplementary Information such that they become part of the publication.

Added statement and reference regarding MATLAB code in the Main text.

Pages 10 Lines 195 – 199 in the Main text

“we use Eqs. 2-5, the average coefficients of $A_\sigma = 0.040$ and $A_E = 2.00$, Vegard’s Law, and the rule-of-mixtures for elastic constants, to search across more than 10,000,000 compositions in the 10-component Al-Cr-Hf-Mo-Nb-Ta-Ti-V-W-Zr family (the Supplementary Note 6, including a MATLAB code, with the data publicly available³⁴).”

34. Francesco Maresca, *et al.* Prediction of yield strength in refractory body-centered-cubic High Entropy Alloys. *Materials Cloud Archive* **2021.65**, (2021) doi: 10.24435/materialscloud:fs-27.

Added paragraphs in the Supplementary Information

Pages 22 – 23 in the Supplementary Information

“The public repository with the raw data and a MATLAB code to predict the yield strengths of BCC HEAs as a function of temperature can be found from <https://archive.materialscloud.org/record/2021.65> and doi: 10.24435/materialscloud:fs-27. The detailed description of the METLAB code and Materials Cloud are shown below.

Prediction of yield strength in refractory body-centered-cubic High Entropy Alloys

How to cite this record

Francesco Maresca, Chanhoo Lee, Rui Feng, Yi Chou, Tamas Ungar, Michael Widom, Jonathan Poplawsky, Yi-Chia Chou, Peter Liaw, William Curtin, Prediction of yield strength in refractory

body-centered-cubic High Entropy Alloys, Materials Cloud Archive 2021.65 (2021), doi: 10.24435/materialscloud:fs-27.

Description

Energy efficiency is motivating the search for new high-temperature metals. Some new body-centered-cubic random multicomponent "high entropy alloys (HEAs)" based on refractory elements (Cr-Mo-Nb-Ta-V-W-Hf-Ti-Zr) possess exceptional strengths at high temperatures, but the physical origins of this outstanding behavior are not known.

Here, by using a recent mechanistic theory, we have computed the high-temperature ($T = 1,300$ K) yield strength based on solute strengthening of over 10 million alloys within the whole Al-Cr-Mo-Nb-Ta-V-W-Hf-Ti-Zr alloy family. Also, the yield strength/density has been computed.

This database enables the efficient search of new alloys with exceptional high-temperature strength.

Materials Cloud sections using this data

No Explore or Discover sections associated with this archive record.

License

Files and data are licensed under the terms of the following license: Creative Commons Attribution 4.0 International.

Metadata, except for email addresses, are licensed under the Creative Commons Attribution Share-Alike 4.0 International license.

External references

Preprint (Preprint where the data is discussed)

F. Maresca, C. Lee, R. Feng, Y. Chou, T. Ungar, M. Widom, K. An, J. Poplawsky, Y.-C. Chou, P. Liaw., W. Curtin, arXiv:2008.11671 (2020)

Journal reference (Paper in which the theory is described)

F. Maresca, W. Curtin, Acta Mater. 182, 235-249 (2020) doi:10.1016/j.actamat.2019.10.015

Keywords

High-entropy alloys, Solute strengthening, High temperature strength, ERC, EPFL”

Comment #5

In conclusion, I strongly recommend this paper for publication in NatComm once the above points have been addressed.

[Reply]

We thank the reviewer for the constructive criticism and for helping to improve the paper and the accessibility of our data. We hope that we have addressed the reviewer’s main concerns in this revised version of the paper.

EDITOR COMMENTS

As you will see, while reviewer #1 raises concerns about discrepancy between the high-throughput prediction model and experimental data, reviewer #3 wants to see better visualization of the screening model presented in Figure 4. Moreover, reviewer #2 raises concerns about discussion of the finding in regards to ref. 15.

[Reply]

We thank all the editor and reviewers for their careful study of our work, and the various comments and questions that have revealed areas where clearer discussion or explanation is needed. We have very carefully revised the manuscript to address all the major comments of all reviewers. Moreover, we have further polished the manuscript to give better understanding for readers.

We believe that the revised manuscript has been significantly improved by going through the careful revisions. Finally, we would like to thank the editor and reviewers for their very helpful and detailed comments and trust that the revised manuscript could be acceptable for publication.

Best regards,

Chanho Lee, Francesco Maresca, Rui Feng, Yi Chou, T. Ungar, Michael Widom, Ke An, Jonathan D. Poplawsky, Yi-Chia Chou, and Peter K. Liaw

Peter K. Liaw

Professor & Ivan Racheff Chair of Excellence

Materials Science and Engineering

406 Ferris Hall

1508 Middle Drive

The University of Tennessee

Knoxville, TN, USA, 37996-2100

Tel: +1 865-974-6356

Fax: +1 865-974-4115

E-mail: pliaw@utk.edu

REVIEWER COMMENTS

Reviewer #1 (Remarks to the Author):

The authors have put a lot of effort in responding to valid concerns raised by three reviewers. There is still a lot of speculation and responses do not conclusively answer the questions raised by reviewers (some are listed below).

1) Authors claim: "Screw dislocations remain important in some BCC HEAs, but the vacancy-driven unpinning of the high-strength screw cross-kinks is expected to lead to a loss of strength at high T". Why does vacancy-driven climb not lead to loss of strength for edge dislocations at high temperature?

2) It is important that a thorough TEM characterization is performed of elevated temperature deformed alloys to understand the dislocation mechanisms. The authors have only shown room temperature ABF-STEM micrographs and it is important to provide dislocation sub-structure evidence that edge dislocations control the strength.

3) After making a claim that edge dislocations control strength of BCC HEAs, the author state that "We did not intend to give the impression that screw dislocations can always be discarded."

Reviewer #2 (Remarks to the Author):

The authors brought answers to all my comments and significantly improved the manuscript to answer the referees' comments. I am therefore in favor of the publication of this article.

Reviewer #3 (Remarks to the Author):

The authors have made a substantial effort to address the concerns raised in the first of review. In particular the figures 4a and S6 related to the screening exercise are now better explained. I now understand that they show strength dependent probability distributions $p(c)$ to find an element at a

certain concentration c , conditional on a strength ranking index. I admit that from the explanation in the original paper I would never have guessed that.

Nevertheless there is still a wide scope for improving presentation. In Figure S6a and S6c the figure and/or caption still give no information that the y axis shows concentration.

Comparing Figures 4 (centre) and S6a, which show in essence the same information, I find some slight differences in presentation. Figure 4 shows shaded areas with thick boundaries, Figure S6a shows shaded areas with three thick lines. Neither the thick boundaries in Figure 4 nor the lines in Figures S6a/c are well explained. My vague *guess* (I shouldn't have to guess) is that in Figures S6a/d the middle thick line shows the average and the upper and lower lines the average plus/minus one standard deviation ('error bars'). But this guess cannot be precisely

correct because the distributions have finite support and at least in some cases, subtracting the standard deviation from the mean will take you to negative concentrations.

The new caption of Figure 4 contains now the enigmatic statement 'the standard deviation (according to a Gaussian distribution) is computed'. I was not aware that the concept of standard deviation and its computation are in any way contingent on the assumption of a Gaussian distribution, which for variables with support on the interval $[0...1]$ is anyway unreasonable.

I suggest to the authors a different way of presentation. Draw a line showing the average concentration as function of the strength index. Then evaluate percentiles of your choice (say, 20percent and 80percent, or 10percent and 90percent) over appropriate windows and create a shaded area encompassed between these percentiles. Explain that the line denotes the mean concentration and the shaded area is the area between the X and y percentiles. Then maybe even the present referee might be able to understand what you are doing.

Other concerns have been adequately addressed and I have no further criticisms or suggestions

REPLY TO REVIEWER'S COMMENTS

We sincerely appreciate the editor's and reviewers' kind/excellent comments on our paper entitled "Strength Can be Controlled by Edge Dislocations in Refractory High-Entropy Alloys". We very carefully considered all points made in their respective comments and have very carefully re-revised the manuscript, as described below:

Reviewer #1 (Remarks to the Author):

Comment #1

The authors have put a lot of effort in responding to valid concerns raised by three reviewers. There is still a lot of speculation and responses do not conclusively answer the questions raised by reviewers (some are listed below).

[Reply]

We truly appreciate the reviewer's rebuttal and acknowledgement of our efforts. We believe that our responses below address the few open questions. We have made very clear the point that edge dislocations CAN control strength in BCC HEAs, by a very detailed experimental/numerical analysis, leaving not much to speculation actually. There are indeed some open issues to be addressed in the future (e.g., the role of vacancy-driven climb), which actually are consequent to this work – one needs to show that edge dislocations can matter for strengthening first, before discussing vacancy-driven climb of edge dislocations and its influence on strengthening. However, these open questions do not detract from the important value of our current findings. As for the

other reviewers, we believe that they have already judged by themselves that their concerns were addressed.

Comment #2

1) Authors claim: “Screw dislocations remain important in some BCC HEAs, but the vacancy-driven unpinning of the high-strength screw cross-kinks is expected to lead to a loss of strength at high T”. Why does vacancy-driven climb not lead to loss of strength for edge dislocations at high temperature?

[Reply]

As detailed in Reference [33] (Maresca and Curtin 2020, Acta Mater 182:144), high-T strength in BCC HEAs is controlled by the cross-kink mechanism, which consists of unpinning of the dislocation from vacancies and interstitials that form along the dislocation line due to cross-kinking (see Fig. 4 in ref. [33], Maresca F, Curtin WA. Theory of screw dislocation strengthening in random BCC alloys from dilute to “High-Entropy” alloys. Acta materialia 182, 144-162 (2020). and related text, as shown below). At high temperatures, where vacancy migration is facilitated, the cross-kink will be spontaneously annihilated. Thus, the cross-kink strengthening of screw dislocations will not occur.

Fig. 4 in Ref. [33]. Formation and atomistic structure of cross-kinks and the associated prismatic loops formed by cross-kink failure. **a** Screw dislocation in a BCC 3-component random alloy (colors indicate species). Arrows indicate the characteristic three-fold symmetric compact core in pure BCC alloys. The dislocation can kink on any $\{110\}$ plane of the $\langle 111 \rangle$ -zone. **b** Kink-pairs formed on different $\{110\}$ planes that, under applied stress, glide and form cross-kinks. **c** Atomistic picture leading to the formation of debris/prismatic loops: two adjacent segments (blue and green) can achieve overall slip on $\{112\}$ (as composition of elementary $\{110\}$ slip processes) following two distinct paths on $\{110\}$ planes. The dashed red box encloses one atomistic (211) plane. A prismatic loop is formed by two adjacent dislocation segments that glide on the distinct paths. **d** Top-view of an atomistic (211) plane showing the atomistic displacements that will eventually lead to formation of a self-interstitial prismatic loop. A self-interstitial (vacancy) forms because the lower (upper) half of the dislocation glides below the atomic plane, displacing the entire atomistic column of atoms (highlighted red arrows) by the Burgers vector \mathbf{b} while the top (bottom) half of the dislocation glides above the atomistic plane, thus leaving the atomic positions unchanged. (For interpretation of the references to colour in this figure legend, the reader is referred to the web version of this article.)

We have not investigated the vacancy-driven climb. This process might indeed influence the high-T strengthening by enabling an extra deformation pathway. However, even with climb, the solute-strengthening due to the random solute distribution will be active. Unlike Ni-based superalloys, there are no easy ways in the BCC HEAs to overcome obstacles since the obstacles in BCC HEAs

are everywhere. Thus, while it is clear that vacancy migration will reduce screw strengthening, it is still unclear what the effect would be on edge strengthening.

Comment #3

2) It is important that a thorough TEM characterization is performed of elevated temperature deformed alloys to understand the dislocation mechanisms. The authors have only shown room temperature ABF-STEM micrographs and it is important to provide dislocation sub-structure evidence that edge dislocations control the strength.

[Reply]

We very much appreciate the reviewer's comments on the characterization of the type of mobile dislocation at high temperatures, using TEM. We state that edge dislocations can control high strength and high-T strength retention. The first part "high strength" is implied to be RT, not high-T. This is what we demonstrate - that edge dislocations are prevalent at RT. The theory then indicates that edge dislocations can give high-T strength retention, and we apply the theory to show this. We agree with the reviewer's suggestions that the quantitative verification of mobile dislocation during high-T deformation is important to support our current claims. Thus, we have conducted the additional characterization of dislocations in NbTaTiV and CrMoNbV alloys via the Bright-Field (BF)-STEM and Stereographic analysis of specimens deformed at $T = 1,173$ K. Figure R1a shows the BF-STEM images of the dislocation networks in NbTaTiV and CrMoNbV, at plastic strains of 11.8 % and 4.2 % at 1,173 K, taken from near the $[113]$ zone axis and the $\vec{g} = (1\bar{1}0)$. All dislocations having a line of length over 5 nm are identified (blue and red lines). The identified line direction is compared with the stereographic projection. If the difference is less than

5 degrees, then the dislocation character is determined. If two characters are possible, the one with the closest match to the dislocation line is chosen. If the dislocation line does not meet these criteria, it is not considered (e.g., the finer-scale dislocation tangles in the images). Figures R1b present the results of the stereographic-projection analysis, which reveals the characters of the long straight dislocation lines. The observed dislocations are highlighted in bold and lie along the lines presented. The $\{110\}\langle 111\rangle$ edge dislocations with $\langle 112\rangle$ dislocation line can be widely observed in NbTaTiV and CrMoNbV. The measured dislocations are predominantly of edge character for both HEAs (27 of 33, or 82 % in NbTaTiV and 81 of 127, or 64 % in CrMoNbV), indicating the maintenance of edge-dislocation-dominant deformation at 1,173 K.

NbTaTiV

CrMoNbV

Figure R1. TEM experiments showing the dominance of edge dislocations in NbTaTiV and CrMoNbV at 1,173 K. **a, b**, Bright-field (BF) STEM image of the 11.8 % - deformed NbTaTiV and 4.2 % - deformed CrMoNbV at 1,173 °C with a two-beam condition near $Z = [113]$ or $Z = [110]$ and the $\vec{g} = (\bar{1}10)$. Inset: HAADF image showing the $\vec{g} = (\bar{1}10)$ plane. Dislocation lines are indicated by red or blue lines corresponding to different angles with respect to the $[1\bar{1}0]$ direction. Due to the high-temperature deformation, many dislocation loops can be observed, which may obscure some dislocation lines. **c, d**, Stereographic projection of possible dislocation-line directions, where $[1\bar{1}0]$ has been aligned with **a** and **b**. The degrees indicate the angle with respect to the $[1\bar{1}0]$ direction. In NbTaTiV, 27 edge and at most 6 screw dislocations are identified. In CrMoNbV, 81 edge and at most 46 screw dislocations are identified.

Regarding the core structure (dislocation sub-structure) of the dislocations, the dislocation cores in NbTaTiV and CrMoNbV are marked in the below figure. Figures R2 (a and c) are the filtered HAADF image of NbTaTiV and CrMoNbV, and the Figs. R2(b and d) are the corresponding Fast Fourier transform (FFT) images. From the HAADF and FFT images, the edge type dislocation with $[111]a/2$ Burgers vector were observed in both NbTaTiV and CrMoNbV. The dislocation core widths were measured, and the value was 1.2 nm and 1.6 nm for NbTaTiV and CrMoNbV, respectively.

Figure R2. The dislocation core structure and diffraction pattern of the strained NbTaTiV and CrMoNbV. **a, c**, the filtered high resolution HAADF image of the 11.8% - strained NbTaTiV and 4.2% - strained CrMoNbV projected along the $[110]$ direction as shown in the corresponded FFT spectrum **b, d**, and the dislocation core was marked as “T”. Based on the FFT spectrum, the extra $(1\bar{1}\bar{1})$ plane can be directly observed, and the corresponding slip plane was on $(1\bar{1}\bar{2})$. The scale bar represented 1 nm.

To address the reviewer’s suggestions, we have added the statements for the core structure of the dislocation and quantitative verification of mobile dislocations during deformation at 1,173

K in the main text and included new TEM images in the Supplementary Information, Fig. S6 and Fig S7, and Supplementary Note 6 and Note 7.

Revised/added statements and figures in the Main Text and Supplementary Information

Page 6, Lines 94 – 96 in the Main text

“The dislocation core structures and diffraction pattern of strained NbTaTiV and CrMoNbV are displayed in Fig S6 in the Supplementary Information (Supplementary Note 6).”

Page 6, Lines 111 – Page 7, Lines 117 in the Main text

“The characterizations of dislocations in the high-T deformed NbTaTiV and CrMoNbV alloys are further investigated via the BF-STEM and Stereographic analysis of specimens deformed at $T = 1,173$ K. (Supplementary Fig. S7) It is found that the $\{110\}\langle 111\rangle$ edge dislocations with $\langle 112\rangle$ dislocation lines can be widely observed in NbTaTiV and CrMoNbV. The measured dislocations are predominantly of edge character for both HEAs (27 of 33, or 82 % in NbTaTiV, and 81 of 127 or 64 % in CrMoNbV), indicating the maintenance of edge-dislocation-dominant deformation at 1,173 K.”

Figure S6. The dislocation core structures and diffraction pattern of the strained NbTaTiV and CrMoNbV. **a, c**, the filtered high-resolution HAADF image of the 11.8 % - strained NbTaTiV and 4.2 % - strained CrMoNbV projected along the [110] direction as shown in the corresponded Fast Fourier transform (FFT) spectrum **b, d**, and the dislocation core was marked as “T”. Based on the FFT spectrum, the extra $(1\bar{1}1)$ plane can be directly observed, and the corresponding slip plane was on $(1\bar{1}2)$. The scale bar represented 1 nm.

Figure S7. TEM experiments showing the dominance of edge dislocations in NbTaTiV and CrMoNbV at 1,173 K. **a, b**, Bright-field (BF) STEM image of the 11.8 % - deformed NbTaTiV and 4.2 % - deformed CrMoNbV at 1,173 K with a two-beam condition near $Z = [113]$ or $Z = [110]$ and the $\vec{g} = (\bar{1}10)$. Inset: HAADF image showing the $\vec{g} = (\bar{1}10)$ plane. Dislocation lines are indicated by red (edge) or blue (screw) lines corresponding to different angles with respect to the $[1\bar{1}0]$ direction. Due to the high-temperature deformation, many dislocation loops can be observed, which may obscure some dislocation lines. **c, d**, Stereographic projection of possible dislocation-line directions, where $[1\bar{1}0]$ has been aligned with **a** and **b**. The degrees indicate the angle with respect to the $[1\bar{1}0]$ direction. In NbTaTiV, 27 edge and at most 6 screw dislocations are identified. In CrMoNbV, 81 edge and at most 46 screw dislocations are identified.

Page 23 in the Supplementary Information

“Note 6. Dislocation core structure of NbTaTiV and CrMoNbV

Figures S6a and S6c shows the atomic-resolution HAADF images of the 11.8 % - strained NbTaTiV and 4.2 % - strained CrMoNbV. Both samples presented the extra $(1\bar{1}1)$ plane as indicated in Figures S6a and S6c with “T”, and the dislocation line was along the viewing direction of $[110]$. It implied that the dislocations had a $(1\bar{1}1)$ Burgers vector and $(1\bar{1}\bar{2})$ slip plane. By the definition of the dislocation-core width, which was the distance between two atomic columns site with $\pm 1/4 b$ Burgers vectors displacements on the extra plane, the corresponding values were 1.2 nm and 1.6 nm for NbTaTiV and CrMoNbV, respectively.”

Page 24 in the Supplementary Information

“Note 7. TEM analysis of dislocation type in high temperature deformed NbTaTiV and CrMoNbV

Figures S7a and S7b present the BF-STEM images of 1,173 K - deformed NbTaTiV and CrMoNbV viewed along the $[113]$. The background was not as clear as in the RT case in Figures 2e and 3e because the high-temperature deformation facilitated the nucleation of a large number of dislocation loops, which may blur some of the dislocation lines. Under the careful examination, the dislocation lines were marked, and the types were determined by aligning the stereographic projection according to the zone axis and the inset HAADF images in Figures 2e and 3e.

For NbTaTiV, the zone axis was $[113]$ and $\vec{g} = (1\bar{1}0)$. The red dislocation lines were edge dislocations, and the blue dislocation lines could be either screw dislocations with a $\langle 111 \rangle$ dislocation line vector or edge dislocations with a $\langle 110 \rangle$ dislocation line vector. Here, we categorized all blue dislocation lines to be screw dislocations, and there were 27 edge and 6 screw

dislocations. The ratio of edge dislocation was 82 %. The actual ratio of edge dislocation could be even higher.

For CrMoNbV, the same method was applied, and the zone axis was $[110]$ and $\vec{g} = (1\bar{1}0)$. Under this condition, the dislocation type can be easily distinguished. There were 81 edge dislocations and 46 screw dislocations. The ratio of edge dislocation was 64%.

The ratio of edge dislocation increased in both NbTaTiV and CrMoNbV at a high temperature, which can further support our claim that edge dislocations played the key role in high-temperatures strengthening.”

Comment #4

3) After making a claim that edge dislocations control strength of BCC HEAs, the author state that “We did not intend to give the impression that screw dislocations can always be discarded.”

[Reply]

The title of our work states clearly that “edge dislocations CAN control strengths of BCC HEAs”. Thus, we did not make the claim that edge dislocations DO control strength of BCC HEAs. Thus, the sentence that we added is to further clarify our point, which is not to claim that screw dislocations can be discarded (this would be indeed wrong), but rather, that edge dislocations CAN be important, which is typically overlooked – as discussed in the previous review and in the paper.

Reviewer #2 (Remarks to the Author):

The authors brought answers to all my comments and significantly improved the manuscript to answer the referees' comments. I am therefore in favor of the publication of this article.

[Reply]

We very much appreciate the reviewer's positive opinions and agreement on our revised manuscript. The re-revised manuscript has been significantly improved by going through the careful re-revisions.

Reviewer #3 (Remarks to the Author):

Comment #1

The authors have made a substantial effort to address the concerns raised in the first of review. In particular the figures 4a and S6 related to the screening exercise are now better explained. I now understand that they show strength dependent probability distributions $p(c)$ to find an element at a certain concentration c , conditional on a strength ranking index. I admit that from the explanation in the original paper I would never have guessed that.

Nevertheless there is still a wide scope for improving presentation. In Figure S6a and S6c the figure and/or caption still give no information that the y axis shows concentration.

[Reply]

We appreciate the careful reading of the manuscript, we have highlighted the information about the y axis in the caption, as shown below.

Figure S8. a, Theory predictions for $T = 1,300$ K strength vs. composition. As detailed in the Supplementary Note 8, computations of yield strengths have been performed for $> 10,000,000$ alloys. The alloys have been assigned with an increasing number as a function of the increasing strength. Thus, the lowest strength alloy is the number 1, and the highest strength alloy is the number 10,003,049. For better visualization, we have grouped the alloys into bins containing 1,000 compositions each. Here, as zoom-in is shown over one bin, which includes the compositions between 9,005,000 and 9,006,000. In the left panel, the shaded areas indicate the compositions between the 10th and 90th percentiles, and the middle line indicates the average of the composition (by considering 1,000 compositions per bin). Thus, the non-dimensional y axis is the elemental concentration. The zoom-in shows the raw data, which are characterized by large fluctuations in the compositions for approximately the same strength. In the example shown, there are 1,000

compositions that can attain ~ 2.13 GPa at $T = 1,300$ K, and all elements of the Nb-Mo-Ta-W-V-Cr-Ti-Zr-Hf-Al compositional space can be used. The elements that are typically in the largest concentrations are Cr, W, and Mo, followed by Zr and Hf. **b**, Theory predictions for $T = 1,300$ K strength/density vs. composition. The plot is constructed in the same way as the panel **a**, by ranking the composition from the lowest to the largest strength/density ratio.

Comment #2

Comparing Figures 4(centre) and S6a, which show in essence the same information, I find some slight differences in presentation. Figure 4 shows shaded areas with thick boundaries, Figure S6a shows shaded areas with three thick lines. Neither the thick boundaries in Figure 4 nor the lines in Figures S6a/c are well explained. My vague *guess* (I shouldn't have to guess) is that in Figures S6a/d the middle thick line shows the average and the upper and lower lines the average plus/minus one standard deviation ('error bars'). But this guess cannot be precisely correct because the distributions have finite support and at least in some cases, subtracting the standard deviation from the mean will take you to negative concentrations.

The new caption of Figure 4 contains now the enigmatic statement 'the standard deviation (according to a Gaussian distribution) is computed'. I was not aware that the concept of standard deviation and its computation are in any way contingent on the assumption of a Gaussian distribution, which for variables with support on the interval $[0...1]$ is anyway unreasonable. I suggest to the authors a different way of presentation. Draw a line showing the average concentration as function of the strength index. Then evaluate percentiles of your choice (say, 20percent and 80percent, or 10percent and 90percent) over appropriate windows and create a

shaded area encompassed between these percentiles. Explain that the line denotes the mean concentration and the shaded area is the area between the X and y percentiles. Then maybe even the present referee might be able to understand what you are doing.

[Reply]

We agree with the reviewer’s remarks, and we have decided to represent the 10 and 90 percentiles in both the Main Text and Supplementary Information (SI). In the SI, which can be more detailed, we provide also the middle average line. We have also included the explanation in the caption. Furthermore, we have also included the MATLAB code as a separate SI file. We hope that this revised version is satisfactory, and we thank the reviewer for helping make the figures clearer to the readers.

Revised/added figure and statements in the Main Text and Supplementary Information

Pages 25 – 26 in the Main text

Figure 4. Theory predictions of yield strengths in BCC HEAs. a, Yield strength vs. temperature, experiments and theory for NbTaTiV and CrMoNbV. *NbTaTiV*: Experiments on the NbTaTiV alloy with 1.15 % O and 0.45 % N: red triangles at a strain rate of 10^{-3} s^{-1} ; theory for an *interstitial-free* alloy: red line. Red square reported in Ref. 9 at $5 \times 10^{-4} \text{ s}^{-1}$ with no O or N impurity content reported. Black circles: experiments on TiHfZrNb at $T = 300 \text{ K}$ with and without 2 % O and N that show strength increase up to $\sim 500 \text{ MPa}^{21}$, comparable to the difference here between model predictions for the interstitial-free NbTaTiV and the experiments with 1.6 % impurities. *CrMoNbV*: Experiments: orange triangles; theory: orange line; no impurities are detected in this alloy. The strength of the new CrMoNbV at 1,173 K exceeds those of all previous reported alloys. **b,** Theory predictions for $T = 1,300 \text{ K}$ strength vs. composition. As detailed in the Supplementary Note 8, computations of yield strengths have been performed for $> 10,000,000$ alloys. The alloys have been assigned with an increasing number as a function of the increasing strength. Thus, the lowest strength alloy is the number 1, and the highest strength alloy is the number 10,003,049. For better visualization, we have grouped the alloys into bins containing 1,000 compositions each (see the Supplementary Note 8). The lowest compositions are for alloys 1 to 1,000 and enter the 1st bin. The second bin contains alloys 1,001 to 2,000, etc. **Within each bin, the interval between the 10th and the 90th percentiles of the elemental contents (Nb, Mo, Ta, etc.) is computed. In this panel, the shaded areas indicate the concentrations between the 10th and the 90th percentiles, evaluated over the compositional bins. Thus, the y axis is the elemental concentration.** The screening includes $> 10,000,000$ compositions in the Nb-Mo-Ta-W-V-Cr-Ti-Zr-Hf-Al compositional space. **c,** A zoom-in of the screening for 100 compositions around the 1,000,000 strongest alloys (between composition numbers of 9,005,000 and 9,005,100). Here, the actual alloy contents per composition

number are shown. There are plenty of possible alloys that can have the same yield strength, but different compositions.

Pages 11 – 12 in the Supplementary Information

Figure S8. a, Theory predictions for $T = 1,300$ K strength vs. composition. As detailed in the Supplementary Note 8, computations of yield strengths have been performed for $> 10,000,000$ alloys. The alloys have been assigned with an increasing number as a function of the increasing strength. Thus, the lowest strength alloy is the number 1, and the highest strength alloy is the number 10,003,049. For better visualization, we have grouped the alloys into bins containing 1,000 compositions each. Here, as zoom-in is shown over one bin, which includes the compositions between 9,005,000 and 9,006,000. In the left panel, the shaded areas indicate the compositions between the 10th and 90th percentiles, and the middle line denotes the average of the composition (by considering 1,000 compositions per bin). Thus, the non-dimensional y axis is the elemental concentration. The zoom-in shows the raw data, which are characterized by large fluctuations in

the compositions for approximately the same strength. In the example presented, there are 1,000 compositions that can attain ~ 2.13 GPa at $T = 1,300$ K, and all elements of the Nb-Mo-Ta-W-V-Cr-Ti-Zr-Hf-Al compositional space can be used. The elements that are typically in the largest concentrations are Cr, W, and Mo, followed by Zr and Hf. **b**, Theory predictions for $T = 1,300$ K strength/density vs. composition. The plot is constructed the same as the panel **a**, by ranking the composition from the lowest to the largest strength/density ratio.

Page 26 in the Supplementary Information

“the 10th and 90th percentiles of the concentrations, per bin, have been computed. The plot reports the shaded area between 10th and 90th percentiles and, in Fig. S8, also the average, per bin, of the concentration of each element.”

Page 27 in the Supplementary Information

“We have also included the MATLAB code as a separate Supplementary Information file.”

Comment #3

Other concerns have been adequately addressed and I have no further criticisms or suggestions.

[Reply]

We thank the reviewer for the careful reading and the very useful advice to clarify the explanation of our data/results!

[Comments, Editor]

Thank you again for submitting your revised manuscript "Strength Can be Controlled by Edge Dislocations in Refractory High-Entropy Alloys" to Nature Communications. We have now received reports from the reviewers who evaluated the original version. On the basis of their comments (copied below), we have decided to invite an additional revision of your work.

You will see that, while the reviewers find that your revisions significantly improved the manuscript, some important points remain to be addressed. We would like you to address the remaining concerns raised by reviewer #1 by possibly providing high-temperature TEM micrographs while addressing the new concerns raised by reviewer #3 to further improve the presentation of the manuscript. Please revise your manuscript, addressing all the remaining issues raised by the reviewers.

[Reply]

We appreciate the editor's kind/excellent suggestions on the re-revision of the manuscript, addressing the reviewer's comments.

We have tried to address the reviewers' respective suggestions and comments by revising/modifying the contents in the Main Text as well as Supporting Information. Moreover, as kindly suggested by Reviewer 1 and the Editor, we have conducted further high-resolution electron microscopy work to identify the characters of dislocations for deformed samples at high temperatures.

We believe that the re-revised manuscript has been significantly improved by going through the careful re-revisions.

Finally, we would like to thank the editor and reviewers for their very helpful and detailed comments and trust that the revised manuscript could be acceptable for publication.

Best regards,

Chanho Lee, Francesco Maresca, Rui Feng, Yi Chou, T. Ungar, Michael Widom, Ke An, Jonathan D. Poplawsky, Yi-Chia Chou, Peter K. Liaw, and William Curtin.

Peter K. Liaw

Professor & Ivan Racheff Chair of Excellence

Materials Science and Engineering

406 Ferris Hall

1508 Middle Drive

The University of Tennessee

Knoxville, TN, USA, 37996-2100

REVIEWERS' COMMENTS

Reviewer #1 (Remarks to the Author):

The authors have attempted to address the reviewers' comments and provided additional TEM results and delineated the scope of this manuscript from following future work.

Reviewer #3 (Remarks to the Author):

My remaining issue regarding the presentation of figures has been satisfactorily addressed. I therefore recommend publication of the manuscript.